# NaLi-H1: A universal synthetic library of humanized nanobodies providing highly functional antibodies and intrabodies

**Sandrine Moutel[1,2,3†], Nicolas Bery[4,5†], Virginie Bernard[1], Laura Keller[4,5,6], Emilie Lemesre[1,2], Ario de Marco[1], Laetitia Ligat[7], Jean-Christophe Rain[8], Gilles Favre[4,5,6], Aurélien Olichon[4,5*], Franck Perez[1,2*]**

[1]Institut Curie, PSL Research University, Paris, France; [2]CNRS UMR144, Paris, France; [3]Translational Research Department, Institut Curie, Paris, France; [4]Inserm, UMR 1037-CRCT, Toulouse, France; [5]Faculté des Sciences Pharmaceutiques, Université Toulouse III-Paul Sabatier, Toulouse, France; [6]Institut Claudius Regaud, Toulouse, France; [7]Le Pôle Technologique du Centre de Recherches en Cancérologie de Toulouse, plateau de protéomique, Toulouse, France; [8]Hybrigenics Service, Paris, France

**Abstract** In vitro selection of antibodies allows to obtain highly functional binders, rapidly and at lower cost. Here, we describe the first fully synthetic phage display library of humanized llama single domain antibody (NaLi-H1: Nanobody Library Humanized 1). Based on a humanized synthetic single domain antibody (hs2dAb) scaffold optimized for intracellular stability, the highly diverse library provides high affinity binders without animal immunization. NaLi-H1 was screened following several selection schemes against various targets (Fluorescent proteins, actin, tubulin, p53, HP1). Conformation antibodies against active RHO GTPase were also obtained. Selected hs2dAb were used in various immunoassays and were often found to be functional intrabodies, enabling tracking or inhibition of endogenous targets. Functionalization of intrabodies allowed specific protein knockdown in living cells. Finally, direct selection against the surface of tumor cells produced hs2dAb directed against tumor-specific antigens further highlighting the potential use of this library for therapeutic applications.

*For correspondence: aurelien. olichon@inserm.fr (AO); Franck. Perez@curie.fr (FP)

[†]These authors contributed equally to this work

## Introduction

Antibodies expended as the biochemical tools of choice to label antigens in cells or tissues. Over the past 20 years, recombinant methods have been developed to quickly select and improve monoclonal antibodies from highly diverse libraries. Recombinant antibodies can be selected from immune or naïve libraries. Immune libraries provide in general high affinity binders but, depending on the antigen, diversity is sometimes limited. Because natural antibody selection requires animal immunization, very conserved or toxic antigens should be avoided and, in general, only limited control of the immune response is possible. On the contrary, non-immune (naïve) libraries provide a higher diversity of binders even for antigens highly conserved in mammals, but high specificity and affinity can be reached only when selecting from very large functional libraries. Immune and naïve libraries are based on the manipulation of antibody fragment that retain binding capacity and specificity of the entire immunoglobulin G (IgG).

The smallest IgG portion capable of binding with high specificity an antigen is the Fv fragment consisting of the variable heavy (VH) and the variable light (VL) domains. In the case of single domain antibodies (sdAb), a VH or a VL alone, is able to bind its target antigen. Each variable domain

**eLife digest** Antibodies are proteins that form part of an animal's immune system and can identify and help eradicate infections. These proteins are also needed at many stages in biological research and represent one of the most promising tools in medical applications, from diagnostics to treatments.

Traditionally, antibodies have been collected from animals that had been previously injected with a target molecule that the antibodies must recognize. An alternative strategy that uses bacteria and bacteria-infecting viruses instead of animals was developed several decades ago and allows researchers to obtain antibodies more quickly. However, the majority of the scientific community view these "in vitro selected antibodies" as inferior to those produced via the more traditional approach.

Moutel, Bery et al. set out to challenge this widespread opinion, using a smaller kind of antibody known as nanobodies. The proteins were originally found in animals like llamas and camels and are now widely used in biological research. One particularly stable nanobody was chosen to form the backbone of the in vitro antibodies, and the DNA that encodes this nanobody was altered to make the protein more similar to human antibodies. Moutel, Bery et al. then changed the DNA sequence further to make billions of different versions of the nanobody, each one slightly different from the next in the region that binds to the target molecules.

Transferring this DNA into bacteria resulted in a library (called the NaLi-H1 library) of bacterial clones that produce the nanobodies displayed at the surface of bacteria-infecting viruses. Moutel, Bery et al. then screened this library against various target molecules, including some from tumor cells, and showed that the fully in vitro selected antibodies worked just as well as natural antibodies in a number of assays. The in vitro antibodies could even be used to track, or inactivate, proteins within living cells.

The NaLi-H1 library will help other researchers obtain new antibodies that bind strongly to their targets. The approaches developed to create the library could also see more people decide to create their own synthetic libraries, which would accelerate the identification of new antibodies in a way that is cheaper and requires fewer experiments to be done using animals. These in vitro selected antibodies could help to advance both fundamental and medical research.

contains four conserved framework regions (FRW or framework) and three regions called CDR (Complementarity Determining Regions) corresponding to hypervariable sequences which determine the specificity for the antigen. VH and VL can be fused together using a synthetic linker and produced as a single protein in the form of a single chain Fv (scFv). Easier to manipulate, they can be produced in several bacteria or eukaryote cell types, fused to various tags or functional domains. Interestingly, antibodies called HCAb in *Camelidae* (*Hamers-Casterman et al., 1993*) or IgNAR in sharks (*Greenberg et al., 1995*) have an antigen recognition part composed of only a VH domain. Camelid natural single domain VH, referred to as VHH or nanobodies, can be expressed as recombinant fragments and represent attractive alternatives over classical antibody fragments like scFvs because they are easy to manipulate and they are not limited by potential misfolding of the two domains (*Wörn and Plückthun, 1999*). It is noteworthy that VHH FRWs show a high sequence and structural homology with human VH domains of family III (*Muyldermans, 2013*) and VHH have comparable immunogenicity as human VH (*Bartunek et al., 2013*; *Holz et al., 2013*). Thus, they further constitute very interesting agents for therapeutic applications, some of them are currently in phase II and Phase III clinical trials (Ablynx Nanobodies; http://clinicaltrials.gov/ct2/results?term=ablynx).

Recombinant antibody fragments allowed not only to accelerate the identification of unique binders, but also the development of a novel type of tool: in this case, the antibodies are directly expressed in living cells as intracellular antibodies (intrabodies), to trace or perturb endogenous target at the protein level. Some scFv or sdAb have indeed been directly expressed in eukaryotic cell as intrabodies to target with high specificity intracellular antigens. Several intrabodies have been used as fluorescent protein fusion to highlight endogenous antigen in cells in a spatio-temporal manner (*Nizak et al., 2003a*; *Rothbauer et al., 2006*). Intrabodies with intrinsic blocking activity have

been reported (*Haque et al., 2011*; *Shin et al., 2005*), and several other approaches have been developed to allow a larger fraction of intrabodies to be used as inhibitory factors: forced co-localization (*Tanaka et al., 2007*), suicide through proteasome targeting (*Joshi et al., 2012*; *Melchionna and Cattaneo, 2007*), rerouting or sequestration to cell compartment (*Böldicke et al., 2005*), degradation (*Caussinus et al., 2011*). Depending on the target, such inhibitors may have potential in human therapy. Production of functional intrabodies depends on the stability of the antibody fragments in the reducing environment of the cytosol that does not allow disulfide bond formation between conserved cysteine. In this context, many advantages of the nanobody scaffold have been reported and, in particular, higher solubility, improved stability in a reducing environment (*Wesolowski et al., 2009*), as well as higher expression yield and thermostability (*Jobling et al., 2003*). For all these reasons, the nanobody scaffold represents an attractive option to generate functional intrabodies.

Thus, we decided to create a non-immune recombinant antibody library of high diversity, based on a nanobody scaffold that would enable efficient in vitro antibody selection against virtually any antigen. Such a library should provide antibodies usable in conventional immune assays and be enriched in antibodies active in the intracellular environment. First, using a fusion assay in *E. coli*, a family of highly functional VHH scaffolds was isolated, optimized for intracellular expression and high stability. One particularly stable VHH scaffold consensus sequence was chosen from these selected antibodies. Additional changes were then introduced to reduce the distance between the *Camelidae* and human VH3 sequences. We confirmed by CDR grafting that this humanized synthetic scaffold (hs2dAb) was robust and functional. Statistics of amino-acid diversity in the CDRs were computed and these information were used to construct a high diversity phage display library of $3.10^9$ independent hs2dAb. The library was then screened against diverse targets of various structures and origin. Highly specific antibodies were selected against EGFP, mCherry, β-tubulin, β-actin, heterochromatin protein HP1α, GTP-bound RHO, p53 and HER2. Affinity measurement indicated that affinities in the nM range can be obtained using this library. As expected from our design, we further showed that hs2dAb are frequently usable as fluorescent intrabodies to track antigens in cells. We also showed that they can be functionalized to induce antigen knockdown. This thus represents the first report of a large and diverse synthetic single domain antibody library enabling fully in vitro selection of highly functional antibodies and intrabodies.

## Results

### Library design

We reasoned that the usual lower quality of these non-immune libraries may come from (1) an antibody scaffold that may not allow robust folding and presentation of CDR region, (2) a lack of control of diversity in the CDR regions and (3) a frequent occurrence of incorrect clones due to the presence of unexpected mutation or empty clones. We designed a pipeline for the development of functional synthetic libraries that aims at overcoming these limitations (*Figure 1A*). As a first step to construct a single domain antibody library enriched in highly stable and functional antibody fragments, we screened for a robust sdAb scaffold (*Figure 1—figure supplement 1*). Previously, we selected from immune or naïve llama VHH libraries several hundreds of clones (*Monegal et al., 2012*; *Olichon and Surrey, 2007*). From this population, we identified a set of robust scaffolds using an assay that discriminates highly stable clones from clones prone to aggregation, or unfolding, in the bacterial cytoplasm (*Olichon and Surrey, 2007*). This assay is based on the fusion of HA-tagged chloramphenicol acetyl transferase (CAT) to the carboxy-terminus of VHH sequences (*Figure 1—figure supplement 1A*). In these conditions, only bacteria expressing a functional VHH fusion in the reducing cytosol (non aggregating, non degraded) can grow in the presence of high antibiotic concentration, thus filtrating a sub-library of potential intrabodies. Expression yield in *E.coli* and apparent solubility as EGFP fusion in the mammalian cell cytoplasm were further assessed to select a set of robust antibody scaffolds. Strikingly, the consensus scaffold was matching the sequence of the most robust VHH framework, represented by a single domain antibody D10 (hereafter named sdAb^D10). When compared to previously reported thermostable nanobodies (*Olichon et al., 2007a*) or intrabodies (*Rothbauer et al., 2006*) obtained from immune libraries, sdAb^D10 was found to provide higher antibiotic resistance in the chloramphenicol filter assay (*Figure 1—figure supplement 1A*). Its expression yield in *E.coli*

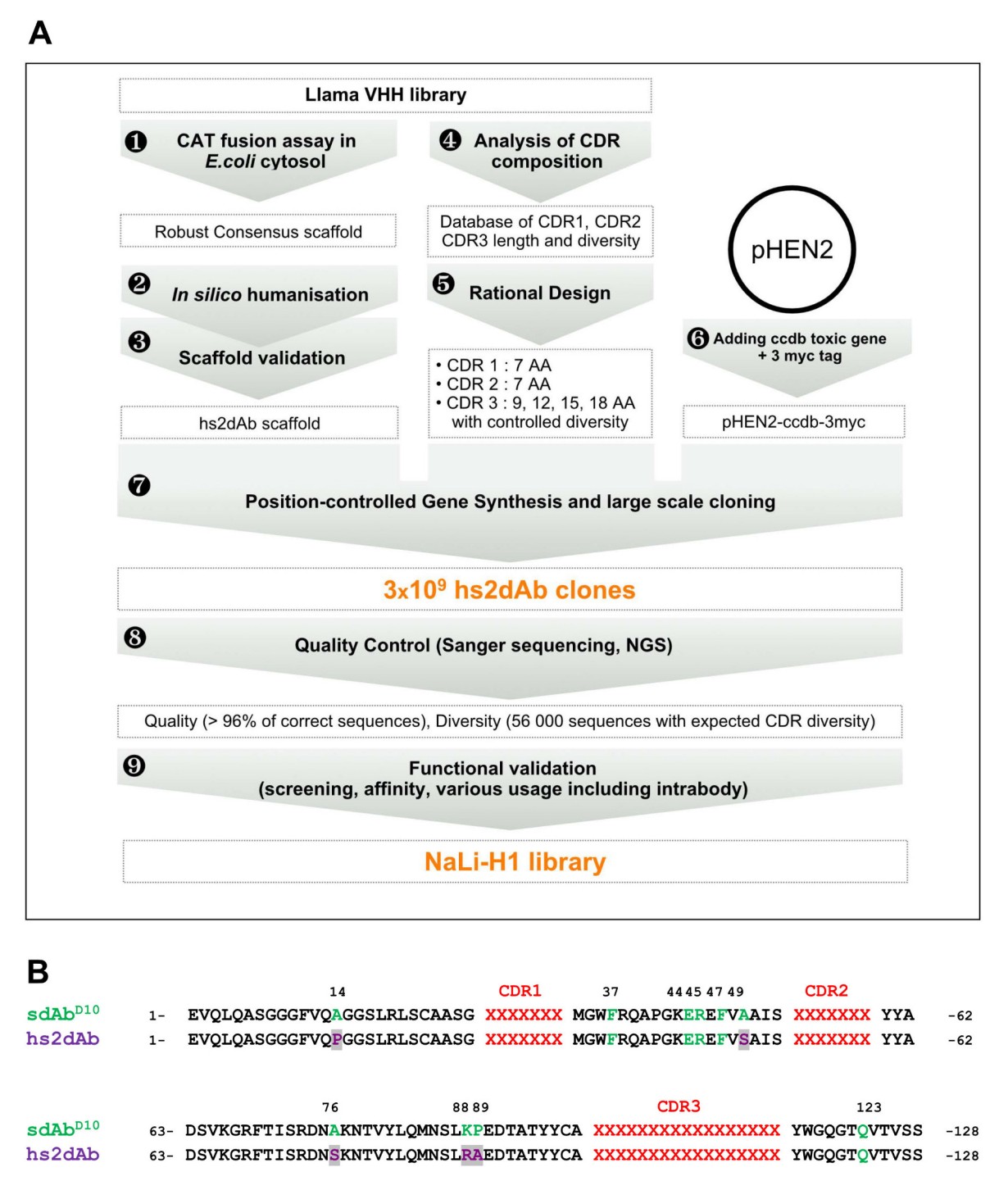

**Figure 1.** Overview of scaffold selection, diversity design, and synthetic production of the NaLi-H1 library. The development of the NaLi-H1 library followed three lines of optimization. (i) A novel scaffold was defined by selecting a set of robust nanobodies using a CAT fusion assay (1). A consensus was derived and mutations were introduced to humanize the scaffold (2). Usability and efficacy of the novel scaffolds (VHH and humanized) were then confirmed evaluating their display on phage, expression in CHO cells and use as intrabodies (3).In silico design was completed analyzing natural CDR diversity (4) and using this information to design synthetic CDRs. A fixed size of 7 aa was chosen for the CDR1 and CDR2. 4 sizes (9, 12, 15 and 18 amino acids) were chosen for CDR3. Finally, the pHEN2 vector was improved by implementing a triple myc tag and inserting a toxic gene (ccdb) to ensure low frequency of empty clones during library construction (6). Gene synthesis (using a tri-nucleotide approach) was ordered, synthetic sequences cloned into the novel pHEN2+ vector, transformed into bacteria and plated on 430 15 cm plates. $3 \times 10^9$ clones were obtained. Quality control was

*Figure 1 continued on next page*

*Figure 1 continued*

carried out using Sanger sequencing of 315 randomly picked clones and large scale sequencing of 56 000 clones. No redundant clone was found. The NaLi-H1 was then screened in various conditions and diversity, efficacy, versatility and affinity evaluated.

The following figure supplements are available for figure 1:

**Figure supplement 1.** Robust scaffold identification.
**Figure supplement 2.** CDR3 loop grafting and synthetic scaffold validation.
**Figure supplement 3.** Nali-H1 library diversity.
**Figure supplement 4.** Solubility of secreted hs2dAb antibodies.

periplasm was in the higher range of soluble llama VHH fragments, allowing efficient and quantitative purification (*Figure 1—figure supplement 1B*). Purified sdAb$^{D10}$ showed excellent solubility, stability after treatment at 70°C and we did not observe aggregation when expressed as an intrabody fused to the EGFP in mammalian cells (*Figure 1—figure supplement 1C*).

To test whether the sdAb$^{D10}$ scaffold composed of the FRW 1 to 4 was robust independently on the CDR sequences, we grafted the CDR loops of the lam1 VHH (*Rothbauer et al., 2006*) directed against laminB into the framework of the sdAb$^{D10}$ (*sdAb$^{D10}$-anti lamin*). In parallel, partial humanization of the scaffold was also tested to figure out whether it affected dramatically its intrinsic properties. Seven residues of the synthetic scaffold sdAb$^{D10}$ were altered towards the most represented in human VH3 while five other divergent residues were kept unchanged. The four llama VHH-specific amino acids hallmarks in the framework-2 region (positions 42, 49, 50, and 52), which are essential to increase intrinsic solubility properties, as well as the Glutamine in position 103, were maintained. We named the resulting hybrid single domain VH hs2dAb (*humanized synthetic single domain antibody*) (*Figure 1B*). After grafting the CDR of the anti-lamin into these two scaffolds, we analyzed their display on the M13 surface (*Figure 1—figure supplement 2A*) and their production by *E. coli* and by CHO cells (*Figure 1—figure supplement 2B*). Both scaffold showed similarly good efficacy. Importantly, both the sdAb$^{D10}$ and hs2dAb scaffolds enabled functional display of the grafted CDRs and robust detection of endogenous lamin was observed by immunofluorescence staining of Hela cells (*Figure 1—figure supplement 2C*). Last, Rothbauer et al. (*Rothbauer et al., 2006*) showed that the original anti-lamin VHH recognized its target antigen upon intracellular expression. Similarly, we observed that both the hs2dAb and the sdAb$^{D10}$ scaffolds allowed the efficient intrabody use of the synthetic anti-lamin antibodies (*Figure 1—figure supplement 2D*). This further indicated that the synthetic scaffolds were robust, non-aggregating and resisted to the reducing conditions found in the cytosol, while allowing the proper display of CDR loops. As partial humanization did not affect the properties of the sdAb$^{D10}$ scaffold, we chose the hs2dAb scaffold as a unique framework to construct a diverse library of synthetic nanobodies endowed with the characteristic stability of these single domain antibodies while displaying an amino acid sequence closer to human VH3.

## Library construction

A synthetic diversity was introduced in the three CDRs by rationally controlling each position of the CDR1 and CDR2 using a set of amino acids that recapitulates partially natural diversity (see the Appendix for more details) while reducing the presence of the most hydrophobic residues in order to avoid the aggregation propensity (see Material and methods). A constant length of 7 amino acids was selected for CDR1 and CDR2. A large spectrum of *Camelidae* VHH CDR3 length is naturally observed and this loop is known to contribute strongly to antigen binding selectivity. Thus, we chose to use four different lengths of CDR3 to cover this spectrum (9, 12, 15 or and 18 amino acids) and introduce random amino acid (except cysteine) at each position.

Synthetic DNA was produced by the trinucleotide DNA assembly and amplification was carried out starting from $2.10^{11}$ different molecules, using only a few cycles of PCR (PCR linearity validated by Q-PCR) to prevent drift during amplification. The synthetic library was inserted into a modified pHEN2 phagemid vector containing a triple myc-tag and suicide gene (ccdB) that allowed positive

selection of insert-bearing clones (*Bernard et al., 1994*). Massive electroporation was carried out using *E. coli* TG1 cells and 430 large agar dishes (140 mm) were used to ensure proper plating of the library. About $3.10^9$ individual recombinant hs2dAb clones were obtained. We named this library NaLi-H1 (for Nanobody Library-Humanized 1). We first evaluated the quality of the NaLi-H1 library by sequencing 315 random clones. Only 13 sequences were found to be incorrect (bearing an in-frame stop codon, missing one base, missing a large region [the CDR1 or CDR1-FWR1- CDR2], or being empty). Thus, a total of about 4% of potential default was observed, which is rather low and only marginal in comparison to the $3.10^9$ clones obtained. The diversity was then evaluated by sequencing $5.6\ 10^5$ inserts using ion Torrent chips (Life Technologies). This confirmed the quality of the library and showed that the four CDR3 lengths (9, 12, 15 or and 18 amino acids) were present in similar proportions. The diversity and statistical distribution of amino acids in the CDRs were found to be as expected (*Figure 1—figure supplement 3*). To estimate the overall folding of the hs2dAb present in the library, we picked randomly 24 clones and tested their solubility in bacteria medium after secretion. Medium were centrifuged at 250 000 *g* and the supernatant and pellet were analyzed. All tested clones showed essentially complete solubility, at least as good as a natural Lama antibody (GFP4). Even after warming at 90°C for 10 min, the hs2dAb showed good solubility (over 70%) (*Figure 1—figure supplement 4*).

## Library screening and validation

The NaLi-H1 library was screened against a set of various antigens. Several standard phage display methods (*Hoogenboom, 2005*) were used (see Materials and methods for details): antigen adsorption on immunotube, native antigen captured on beads, direct selection at the cell surface. All conditions allowed the efficient recovery of diverse and functional antibodies.

As a first screen to evaluate the quality of the library, we chose to select specific binders for the EGFP and mCherry fluorescent proteins. NaLi-H1 phages were panned against biotinylated EGFP or mCherry and 3 rounds of selection were carried out. Eighty clones were analyzed for each screening campaign. From the panning against EGFP, 37 non redundant nanobodies were shown to detect EGFP by phage ELISA. These antibodies were then used for immunofluorescence and 10 of them were found to efficiently stain EGFP in fixed HeLa cells (*Figure 2A*). Similarly, selection against mCherry led to 6 positive binders (*Figure 2B*). As shown in *Figure 2A and B*, no staining was obtained in untransfected cells.

In the next screen, two highly and constitutively expressed components of cell cytoskeleton, tubulin and β-actin, were targeted. Antibodies against tubulin were selected in native conditions (*Nizak et al., 2003a*) using commercial biotinylated tubulin (Cytoskeleton). After two rounds of selection, 3 out of 40 clones analyzed were shown to detect endogenous tubulin by immunofluorescence (*Figure 2C*). As expected, staining was lost in cells treated with the microtubule destabilizing drug nocodazole, with only a few stable microtubules being labelled in these conditions (*Figure 2— figure supplement 1A*). This antibody was recently used to stain microtubules by super-resolution imaging (*Mikhaylova et al., 2015*). Among phage display methods, selection on antigens directly adsorbed on the surface of immunotube is often used as a cheap and straightforward method, despite the low capacity and the strong denaturation imposed by non-specific adsorption. A screen against coated β-actin led to the identification of 16 unique binders positive in phage ELISA. Seven of these antibodies decorated endogenous actin stress fibers by immunofluorescence in MRC5 cells (*Figure 2D*) as well as in other cell lines. Treatment of cells with cytochalasin D disorganizes actin fibers. Accordingly, staining with the hs2dAb was strongly altered (*Figure 2—figure supplement 1B*). Three of the hs2dAb detected a single band at the molecular weight of β-actin in western blot from HeLa cell extract while one of the anti-tubulin detected a band at the correct molecular weight of tubulin (*Figure 2—figure supplement 1C*).

Actin and tubulin are strongly expressed cellular proteins. Another screen was thus performed to select binders directed against proteins expressed at a lower level. A first screen was carried out against the tumor suppressor p53 protein. The 83 first amino acids of the NP_000537.3 isoform were produced in bacteria fused to a SNAP tag, biotinylated in vitro and used as a target in the phage display selection. Among 12 clones positive in phage ELISA, 6 selected hs2dAb were shown to label endogenous p53 in immunofluorescence on A431 cells (*Figure 2E*). The specificity of the staining was confirmed using RPE-1 cells (*Figure 2—figure supplement 2*). A low nuclear staining was observed in normal conditions, with some variability between cells. As expected, the intensity was enhanced

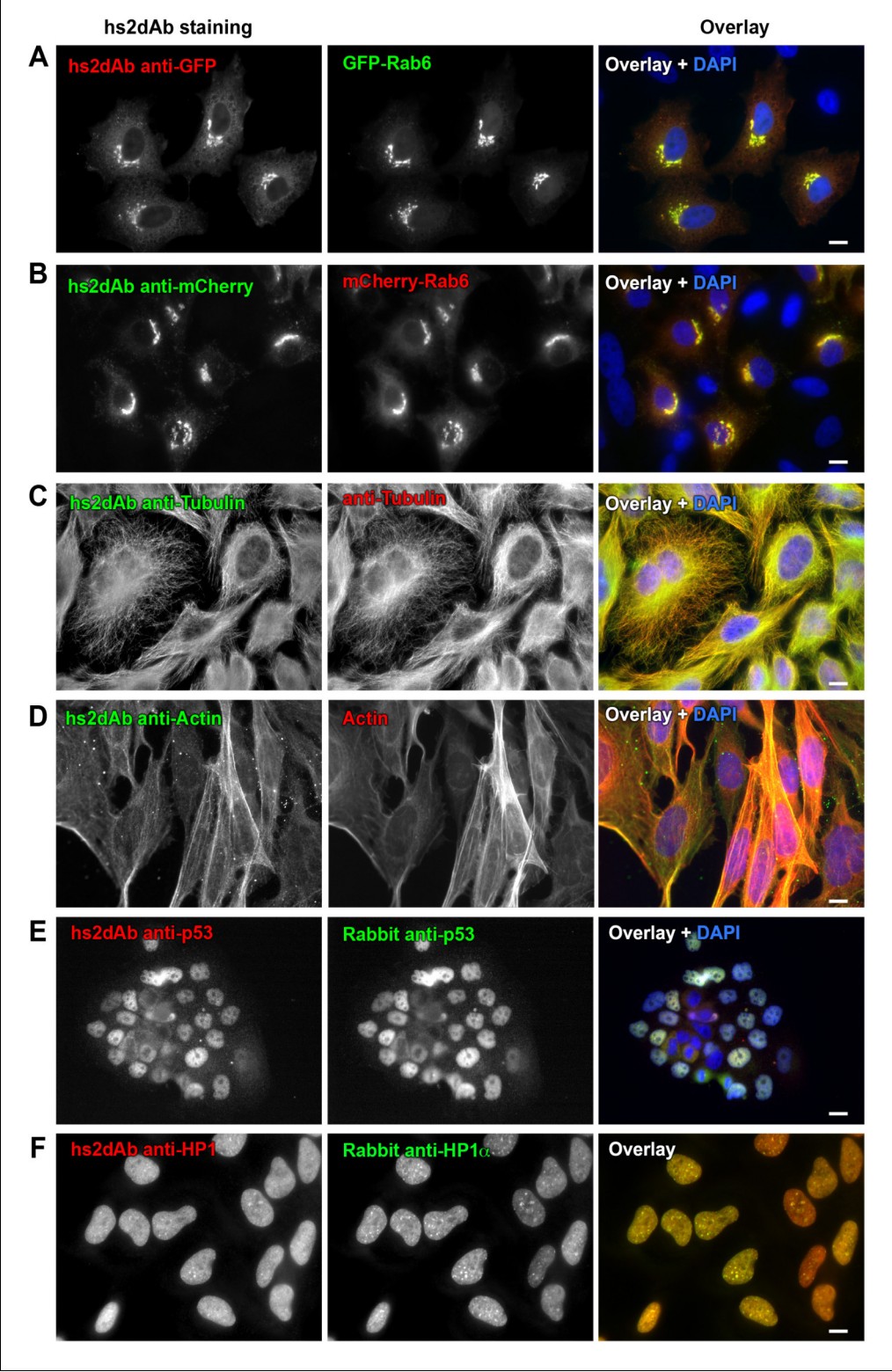

**Figure 2.** Selection of functional hs2dAb against various antigens. (**A**) HeLa cells were transfected with GFP-Rab6, fixed using paraformaldehyde, permeabilized using saponin and stained with non-purified myc-tagged hs2dAb (R3TF3) directed against EGFP and revealed with anti-Myc-Tag (9E10) and Cy3-labeled secondary antibodies. (**B**) HeLa cells transfected with mCherry-Rab6, fixed and permeabilized as in A and stained using a myc-tagged non purified hs2dAb against mCherry (C11). The hs2dAb was then revealed using 9E10 and A488-labeled secondary

*Figure 2 continued on next page*

*Figure 2 continued*

antibodies. (**C**) Cells were fixed in methanol and co-stained with a non-purified anti-tubulin hs2dAb (D5) fused to a human Fc domain and a mouse monoclonal anti-tubulin antibody (DM1A), and revealed using an anti-Human Fc-A488 and an anti-Mouse-Cy3 secondary antibody, respectively. (**D**) hs2dAb F4 anti-beta-actin was used to stain MRC5 cells fixed with paraformaldehyde and post fixed with methanol. The hs2dAb was then revealed using 9E10 and A488-labeled secondary antibodies. Cells were co-stained by red fluorescent phalloidin to detect actin stress fibers. (**E**) A431 cells were fixed with 3% paraformaldehyde, permeabilized with 0.1% Triton and stained with the anti-p53 hs2dAb (B7) fused to a human Fc domain together with a rabbit polyclonal antibody directed against p53. Immuno-labeling was revealed using anti-Human Fc-Cy3 and anti-Rabbit-A488 secondary antibodies. (**F**) The hs2dAb antibody directed against HP1α (A5) fused to a human Fc domain was used to stain HeLa cells fixed with paraformaldehyde and permeabilized with 0.1% TritonX100. Cells were co-stained using a polyclonal rabbit antibody directed against HP1α and immuno-labeling was revealed using anti-Human Fc-Cy3 and anti-Rabbit-A488 secondary antibodies. (scale bar = 10 µm).
The following figure supplements are available for figure 2:

**Figure supplement 1.** Specificity of hs2dAb directed against tubulin and actin.

**Figure supplement 2.** Specificity of the anti-p53 hs2dAb.

**Figure supplement 3.** Specificity of the anti-HP1 hs2dAb.

upon UV-induced DNA damage. Such an increase was not observed in a RPE-1 cell line stably expressing an shRNA against p53. A second screen was carried out against the heterochromatin protein HP1α. HP1α was produced in bacteria fused to an avitag to obtain a biotinylated recombinant protein. Biot-HP1α was then immobilized on streptavidin beads and used as a target for 3 rounds of selection. 5 individual hs2dAb were directly identified by immunofluorescence staining of HeLa cells. *Figure 2F* shows that selected antibodies were efficiently staining endogenous HP1 in the nucleus. Overexpression of the different HP1 variant followed by western blot and immunofluorescence analysis actually suggested that HP1β/γ were also detected by the antibody (*Figure 2—figure supplement 3*). Together these results showed that the NaLi-H1 synthetic library can be screened in various conditions against very different purified targets while leading to the rapid identification of diverse and specific binders that can be used in classical antibody-based staining methods.

## Selection of conformation-sensitive antibodies

One of the main advantages of full in vitro immunization using display technologies is the control of antigen conformation and concentration. It allows to drive selection towards the desired outcome. For example, selection schemes can be devised to improve the recovery of high affinity binders endowed with low off-rate kinetics (*Lee et al., 2007*), to target specific epitopes (*Even-Desrumeaux et al., 2014*; *Vielemeyer et al., 2009*), or to identify conformation sensitive-binders (*Haque et al., 2011*). Recombinant antibody fragment library screening have, for example, provided several binders targeting selectively the active conformation of GTP binding proteins (*Dimitrov et al., 2008*; *Nizak et al., 2003a*; *Tanaka et al., 2007*). We hypothesized that the NaLi-H1 synthetic library had enough diversity and functionality to enable the identification of selective conformational binders. A subtractive panning was performed to select conformation-specific antibodies directed against small GTPases from the RHO subfamily (*Chinestra et al., 2012*). Small GTPases are molecular switch that cycle between an inactive and an active state when bound to GDP or GTP nucleotides, respectively. Mutant of small GTPases can be designed that adopt stably an active or inactive conformation. A constitutively active (CA) mutant RHOA L63 was expressed in HEK293 as a bait then freshly pulled down for panning to preserve its native conformation. To enrich in phage specific for GTP-bound RHOA, a depletion step was introduced from the second round of panning using GDP-bound RHO proteins. After four rounds of selection, clones were analyzed using phage ELISA against either wild type RHOA loaded with GTPγS (a non-hydrolysable analogue of GTP) or GDP-loaded RHOA. Forty clones presenting a differential ELISA signal in favor of the GTP loaded RHOA were sequenced. One antibody, represented by clone H12, represented more than 50% of the population. We analyzed H12

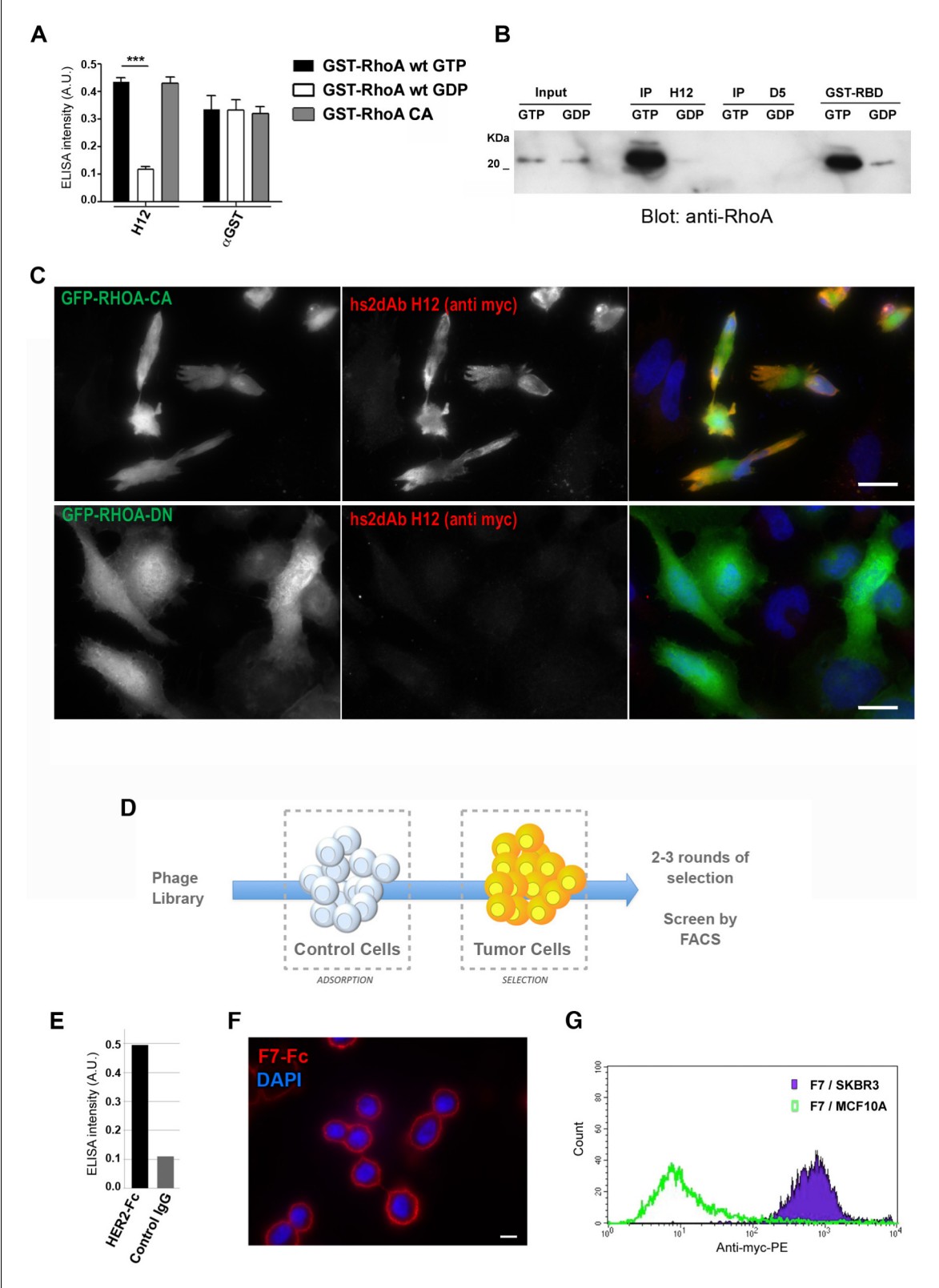

**Figure 3.** Subtractive selection led to conformational or cell type specific hs2dAbs. (**A–C**) H12 is a conformational hs2dAb binding only to the GTP bound, activated state, of the RHOA GTPase: (**A**) ELISA using the H12 or anti-GST antibodies to reveal recombinant GST-RHOA wild type proteins loaded with either 100 μM GTP gamma S (Black) or 1 mM GDP (White), or constitutively active mutant proteins GST-RHOA Q63L (Grey). Means ± SEM. (**B**) A CBD tagged H12 pull down from HeLa cell extract loaded with100 μM GTP gamma S (GTP) or with 1 mM GDP as inputs. Western blot reveals

*Figure 3 continued on next page*

*Figure 3 continued*

RHOA at a similar level in 5% of both input but only on the GTP loaded extract in the CBD-H12 pull down. D5 anti tubulin was used as a negative control and the standard GST-RBD (RHO binding domain of Rhotekin) as a positive control of active RHO pull down. (C) Immunofluorescence on HeLa cells overexpressing GFP-RHOA CA (constitutively active) mutant or GFP-RHOA DN dominant negative mutant. H12 staining detected using a myc tag antibody revealed only cells overexpressing the constitutively active mutant with a pattern stronger at the cell periphery were RHOA activation is high. (D) Tumor cell surface subtractive selection scheme. (E) ELISA of hs2dAb F7 anti-HER2 on HER2 fused with a rabbit Fc versus binding on rabbit Fc at equimolar concentration. (F) hs2dAb F7 anti-HER2 decorated the SKBR3 membrane in immunofluorescence. SKBR3 cells were fixed with 3% paraformaldehyde and stained with F7 revealed by an anti-HisTag (Sigma) and an anti-MouseCy3 secondary antibody (Jackson). (G) FACS analysis of F7 anti-HER2 on SKBR3 HER2 positive cells versus MCF10A HER2 negative cells. (Scale bar = 10 μm).

The following figure supplement is available for figure 3:

**Figure supplement 1.** non cropped western blot corresponding to *Figure 3B* detection RHOA.

binding specificity by ELISA on several purified RHO proteins expressed as GST fusion in *E.coli*. H12 recognized the constitutively active mutant RHOA L63 (RHOA-CA) which is bound to GTP due to impaired hydrolysis activity. A similar signal was obtained with wild type RHOA loaded with the non-hydrolysable GTP analogue GTPγS. In contrast, no binding was observed to the dominant negative RHOA N19 mutant RHOA-DN nor to GDP-loaded wild type RHOA (*Figure 3A*). The capacity of H12 to specifically immunoprecipitate GTP-loaded RHOA from mammalian cell extracts was then evaluated in comparison to the standard method to assay RHO activity (*Ren et al., 1999*). This pull down method is based on the RHO binding domain of Rhotekin fused to GST (GST-RBD) which is known to bind to the active conformation of RHO GTPase. The hs2dAb H12 bearing a carboxy-terminal CBD (Chitin-Binding Domain) was expressed in *E. coli* and immobilized on chitin beads. These beads were then incubated with HeLa cell extracts pre-treated with either GTPγS or GDP to load small GTPases with the respective nucleotide. The H12 hs2dAb was found to be highly selective of RHO loaded with GTPγS as it was unable to precipitate RHO from GDP loaded extract (*Figure 3B*). A similarly strong conformation-specificity was found when using H12 for immunofluorescence staining (*Figure 3C*). HeLa cells expressing the GFP-RHOA constitutively active mutant carrying the mutation Q63L (GFP-RHOA CA) or the dominant negative mutated T19N (GFP-RHOA DN), were fixed and stained with H12 hs2dAb. Expression of the dominant negative mutant GFP-RHOA DN, did not lead to an increased signal over the background of un-transfected cells. In contrast, a staining with H12 was correlated with the level of GFP-RHOA CA mutant expression. Note that the signal does not fully overlap GFP fluorescence and appeared stronger at the cell border and in large zone where cell shape is strongly retracted by large bundled actin stress fibers induced by a sustained activation of the RHOA/ROCK pathway (*Mayer et al., 1999*). This CA mutant, like active mutant of many small GTPases related to RAS still need to be activated by guanine nucleotide exchange factors to be loaded with GTP and display the active conformation. Thus, we believe that the H12 staining revealed the active form of this mutant in cells (*Figure 3C*). All together, these results demonstrated that the H12 hs2dAb is selective for RHO GTPases in their active conformation, highlighting the performance and diversity of the NaLi-H1 library.

## Direct selection against cell surface antigens

The use of antibodies to target cells involved in pathologies like cancers or viral infection is one of the most promising therapeutic approach. Antibodies also represent a unique tool to identify novel targets at the cell surface. Using synthetic libraries like the NaLi-H1 library to carry out direct selection against the cell surface of tumor cells would strongly accelerate the identification of specific antigens while allowing extended control over the conditions of selection. A subtractive selection scheme was set up to identify antibodies selectively detecting the surface of breast tumor cells: phages displaying hs2dAbs were first depleted against a reference cell line before being selected against the target one (*Figure 3D*). As a target cell line, we used the SKBR3 line, which is known to overexpress the HER2 cell surface protein, while the MCF7 cell line, negative for HER2, was used to pre-adsorb the library. After the third round of bio-panning, 88 clones were analyzed by FACS and 58 were found to be positive when tested on SKBR3 cells and negative on MCF7 cells. Sequencing the 58 positive clones revealed that 15 independent binders had been selected. Although the

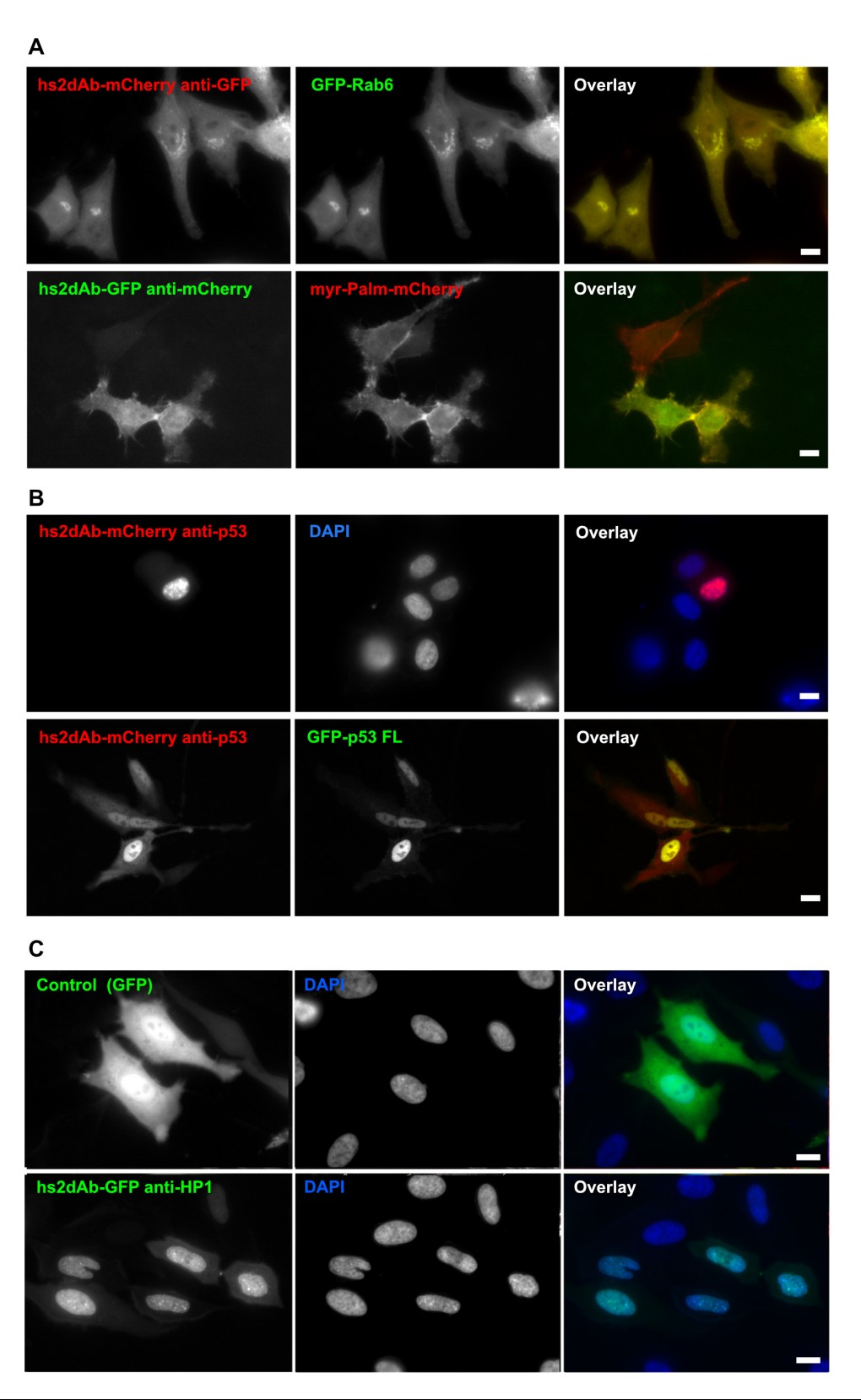

**Figure 4.** Fluorescent intrabodies tracking endogenous proteins. Intracellular expression of hs2dAb. (**A**) (top panel) HeLa cells were co-transfected with GFP-Rab6 and a hs2dAb-mCherry anti-EGFP plasmids. The hs2dAb
*Figure 4 continued on next page*

*Figure 4 continued*

mCherry anti-EGFP colocalized perfectly with the Rab6 Golgi staining. (bottom panel) HeLa cells were co-transfected with Myr-palm-mCherry and a VHH-EGFP anti-mCherry plasmids. The VHH-EGFP anti-mCherry interacted with its target in vivo and colocalized perfectly with the mCherry staining at the plasma membrane. (B) SKBR3 cells were transfected with an anti-p53 hs2dAb-mCherry alone (top panel), or together with full length p53-EGFP which concentrated the hs2dAb into the nucleus (bottom panel). (C) GFP, used as a control (top panel) or a GFP-tagged anti-HP1 hs2dAb (bottom panel) were transiently expressed in HeLa cells (green). In contrast to the GFP control, the GFP-tagged anti-HP1 strongly accumulated in the nucleus where it labeled nuclear condensations. (Scale bar = 10 µm)

subtractive selection was not performed on identical cell lines expressing or not HER2, we observed strikingly that 12 clones out of 15 recognized HER2 exoplasmic domain by ELISA using HER2-Fc as a target antigen (*Figure 3E*), suggesting that it behaves as a dominant differential epitope. As shown in *Figure 4*, these antibodies efficiently detected HER2 at the cell surface by immunofluorescence (*Figure 3F*) or by FACS (*Figure 3G*). These experiments demonstrated that the NaLi-H1 library will represent a unique tool to discover, in a rapid and cost effective manner, specific antibodies detecting antigens present at the surface of pathological cells. These antibodies may then be used to identify the corresponding target.

## hs2dAbs as intracellular antibodies

Various antibody fragments have long been proposed to represent powerful tools when expressed in cells as intrabodies. Although several studies indeed report efficient use of intrabodies (reviewed in [*Kaiser et al., 2014*; *Lobato and Rabbitts, 2003*; *Stocks, 2005*]), this is limited to antibody scaffolds that resist to the reducing environment of the cytosol. We evaluated the use of the hs2dAb scaffold to develop intrabodies. Randomly chosen hs2dAbs were fused to a fluorescent protein and observed in living cells. In comparison to our previous experience with scFv or nanobody libraries in which a majority of tested antibodies showed aggregation when expressed in the cytosol as an EGFP-fusion, most of the hs2dAbs tested here gave a monodispersed fluorescence, and very few were showing aggregates. Monodispersed fluorescence of the EGFP has been proposed to be a convenient indicator of stability and potential use as intrabody (*Guglielmi et al., 2011*). Several hs2dAb directed against actin, tubulin, EGFP or p53 were tested for their ability to trace intracellular antigens in living cells. None of the anti-tubulin or anti-actin antibodies tested were found localized on microtubule or microfilament, respectively, in living cells. We reasoned that this poor efficiency may be linked to the antigen denaturing conditions used during the selection of antibodies directed against actin. The existence of a large pool of unpolymerized actin and tubulin may also prevent efficient recruitment on the polymers. In contrast, several anti-GFP and anti-mCherry hs2dAbs were found to efficiently label their targets, like for example GFP-Rab6 or Myr-Palm-mCherry fusion proteins, in living cells (*Figure 4A*). Similarly, the anti-p53 hs2dAb fused to mCherry were clearly accumulated in the nucleus of SKBR3 cells where endogenous p53 is also localized. This signal was enhanced in cells overexpressing p53-EGFP (*Figure 4B*). This further confirmed the binding specificity of the hs2dAb anti-p53 while expressed in the reducing cytosol. Effective intrabodies were also obtained against HP1. *Figure 4C* (lower panel) shows that anti-HP1 expressed as an EGFP tagged protein in HeLa cells localized in the nucleus where it labels condensed structures similar to HP1 usual staining. In contrast, a diffuse cytoplasmic and nuclear staining was obtained using non relevant hs2dAbs fused to EGFP (*Figure 4C*, upper panel).

These results indicate that functional intrabodies can be obtained at high frequency using the NaLi-H1 library. Intrabodies can be used, upon fusion with a fluorescent protein, to track the dynamics of their target in living cells (see for example *Nizak et al., 2003a*). Such an

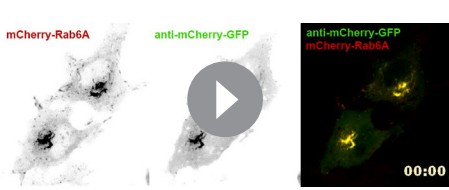

**Video 1.** mcherry-Rab6 was transiently expressed in HeLa cells together with an anti-mCherry hs2dAb fused to GFP. 24 hr after transfection, cells were imaged using a spinning disk confocal microscope.

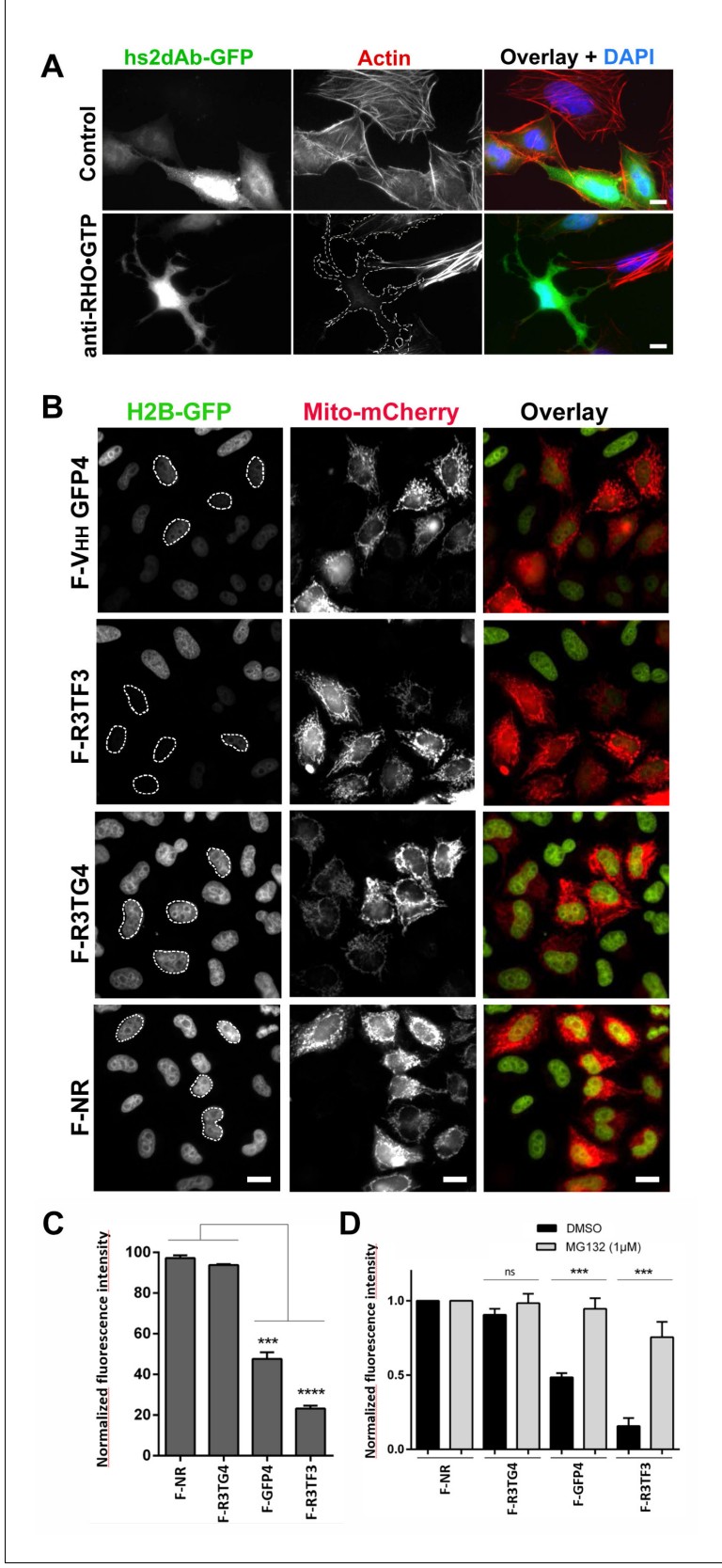

**Figure 5.** Targeting cellular proteins using inhibitory antibodies or by functionalizing antibodies to induce protein knockdown. (**A**) HeLa cells expressing transiently a EGFP-tagged non relevant hs2dAB (top panel) or EGFP-H12

*Figure 5 continued on next page*

*Figure 5 continued*

anti RHO-GTP (bottom panel) were fixed 20 hr post transfection and stained using DAPI and Alexa 594 phalloidin to detect actin stress fibers. The H12 hs2dAb induced actin stress fibers disappearance and major cell shape change (see cells outlined with a dotted line) (**B**) Protein knockdown of H2B-EGFP mediated by functionalized inhibitory antibodies. HeLa S3 cell stably expressing histone H2B-EGFP were transfected with vectors expressing antibodies fused to an F-box (F-Ib) to induce degradation of the targeted cellular antigen. The F-GFP4 VHH (DegradFP) was used as a positive control (top panel) and a non-relevant hs2dAb as a negative control (bottom panel). F-Ib were expressed using a bi-cistronic vector driving the co-expression of mitochondrial targeted mCherry. Protein interference is analyzed in cells displaying mCherry positive mitochondria (mitoCherry channel). Efficient protein knockdown is obtained using the R3TF3 anti-EGFP intrabody. Note that not all nanobodies can be used as F-Ib because R3TG4 does not induce protein degradation. (Scale bars = 20 µm) (**C**) Fluorescence decay measurement of the protein interference assay was quantified by flow cytometry (10000 cells analyzed, from 3 independent replicates). GFP fluorescence intensity was quantified in the transfected and the untransfected subpopulations for each F-Ib. The ratio of each median of fluorescence (transfected versus untransfected population) was calculated as a percentage of GFP fluorescence intensity for one F-Ib. A strong decrease in fluorescence corresponding to protein knockdown was observed with F-GFP4 VHH and F-R3TF3 hs2dAb intrabodies while the non-relevant negative control and R3TG4 did not induce a decrease of fluorescence. (**D**) Cells were analyzed as in C but the cells were incubated in 1 µM MG132 or DMSO for 44 hr after transfection by the different F-Ib and fluorescence intensity was normalized using the non-relevant control. Protein knockdown was inhibited by MG132.

The following figure supplements are available for figure 5:

**Figure supplement 1.** Conformational selectivity of the H12 intrabody towards RHOA.GTP.

**Figure supplement 2.** Protein knockdown set up using F-Ib degradation with anti-GFP intrabodies.

**Figure supplement 3.** non cropped western blot corresponding to *Figure 5—figure supplement 1* detection of RHOA, Myc tagged hs2dAb intrabodies, and GAPDH which is not in the main figure.

**Figure supplement 4.** non cropped western blot corresponding to *Figure 5—figure supplement 1D* detection of tubulin, GFP and myc tag.

---

application is illustrated in the *Video 1* where an hs2dAb directed against mCherry is used to track mCherry-fused Rab6 in living cells. Intrabodies may allow not only to track the dynamics of their cellular target, but also to perturb, or block, their activity. Our results indeed indicate that the H12 antibody was able to perturb endogenous RHO activity when expressed in the cytosol. We could not directly image enrichment of the H12 antibodies in cellular sub-domains in living cells but we observed that H12 may behave as an efficient intrabody carrying out co-immunoprecipitation experiments. H12 carrying a carboxy-terminal myc tag was expressed in HeLa cells together with either the CBD-fused RHOA DN or with the CBD-RHOA CA mutants. H12 was pulled-down by RHOA CA but not by inactive RHOA DN (*Figure 5—figure supplement 1*). H12 thus worked as an intrabody and kept its conformation sensitivity in the cytosol. Because RHO GTPases are involved in signaling pathways that promotes the actin cytoskeleton polymerization, we looked at functional effects induced by H12 overexpression. In contrast to un-transfected cells or cells transfected with various non-relevant EGFP fused hs2dAb (*Figure 5A*, upper panel), we observed that cells expressing H12-EGFP were totally devoid of actin stress fibers (*Figure 5A*, lower panel). This alteration in actin filament organization was associated with marked changed in cell shape characteristic of loss of intracellular mechanical forces and tension. As RHOA plays a major role in activating myosin II and actin cytoskeleton reorganization, our results suggested that H12 efficiently perturbed RHO-dependent signaling, mimicking the phenotype induced by the C3 exoenzyme RHO inhibitor (*Ridley and Hall, 1992*).

Identification of blocking antibodies is a challenging task and not all functional intrabodies are inhibitory. However, it is possible to functionalize non-blocking intrabodies to inhibit their target function. One approach relies on the ubiquitinylation and degradation of the recognized target as described by *Caussinus et al. (2011)*. This approach is based on the fusion of intrabodies to an F-box domain which allows interaction with Skip1, a member of the SCF complex, an E3 ubiquitin

ligase of the complex E1/E2/E3 ubiquitinylation machinery, that targets proteins to proteasome-dependent cellular degradation (*Caussinus et al., 2011*) (*Figure 5—figure supplement 2*). This approach was efficiently developed to target several EGFP fusion proteins in *Drosophila* using a single anti EGFP intrabody, named GFP4, which is a robust and high affinity EGFP llama intrabody originally isolated from an immune library (*Rothbauer et al., 2006*). To get insight into the relative functionality of hs2dAb for such a protein interference approach, several of the anti EGFP hs2dAb selected from the NaLi-H1 library were fused at their amino terminus to the Fbox domain and their efficacy was compared to the efficacy of the Fbox-GFP4 nanobody. To detect cells expressing Fbox-intrabody fusion proteins (F-Ib), we constructed a bicistronic vector driving the co-expression of F-Ib together with a mitochondria-targeted mCherry (Mito-mCherry) (*Figure 5B*). We expressed the F-Ib antibodies in a HeLa clone stably expressing EGFP fused to histone H2B (*Silljé et al., 2006*) and looked for EGFP-H2B depletion. As expected, F-GFP4, also known as degradFP, induced a strong reduction of H2B-EGFP expression as analyzed by western blot (*Figure 5—figure supplement 2*). Accordingly, a strong reduction in nuclear fluorescence intensity was observed in cells expressing F-GFP4 (see mito-mCherry positive cells, *Figure 5B*; *Figure 5—figure supplement 2*). No effect was observed when expressing either GFP4 alone or a GFP4 fused to a truncated, nonfunctional, Fbox domain (*Figure 5—figure supplement 2*). When anti-EGFP clones selected from the NaLi-H1 library were tested, we observed that some of the hs2dAb that were found to active as fluorescent intrabodies failed to degrade H2B-EGFP when expressed as F-Ib. This highlights the fact that not all intrabodies can efficiently be functionalized with the F-box and that in vivo binding to a target is not the only parameter to consider. However, several hs2dAb anti-EGFP induced a complete disappearance of nuclear H2B-EGFP signal when expressed as F-Ib (F-R3TF3, *Figure 5B*) while no reduction was observed when using an anti-EGFP that cannot be used as an intrabody (F-R3TG4, *Figure 5B*). FACS analysis showed a decreased of fluorescence intensity by as much as 70% (*Figure 5C*). As expected, this effect was reversed in the presence of a proteasome inhibitor (*Figure 5D*). Altogether, these experiments show that the hs2dAb scaffold enables the frequent selection of antibodies that can be expressed in the mammalian cell cytoplasm to be used as functional fluorescent or inhibitory intrabodies.

## Discussion

Here, we report the construction of the first large fully synthetic single domain antibody library based on a humanized scaffold derived from llama VHH. A set of robust nanobody scaffolds was first identified using a positive expression screening in *E. coli* cytosol. One very robust scaffold (sdAb$^{D10}$) was identified and was used as a base. After introduction of several modifications that aimed at humanizing its primary sequence, we designed the hs2dAb scaffold which is as stable as sdAb$^{D10}$ while being closer to human VH3. Our data indicate that the hs2dAb displays partial resistance and/or refolding after treatment for 10 min at 90°C. Using CDR grafting experiments we confirmed the efficacy and the stability of the synthetic scaffold to display CDR regions. Based on our prior experience on phage display libraries, immune or naïve llama VHH libraries (*Monegal et al., 2012*; *Olichon and Surrey, 2007*) or from scFv libraries (*Dimitrov et al., 2008*; *Goffinet et al., 2008*;

**Table 1.** Summary of screenings showing the number of unique clones giving positive signal. (ND means non determined)

| Antigen | Phage ELISA | IF/FACS | Intrabody | Rounds of panning |
|---|---|---|---|---|
| | | **Positive clones** | | |
| GFP | 37 | 10/ND | 4/10 | 2 |
| mCherry | ND | 6/ND | 2/6 | 3 |
| Tubulin | ND | 3/ND | 0/3 | 2 |
| Actin | 16 | 7/ND | 1/7 | 3 |
| p53 | 12 | 6/ND | 2/6 | 2 |
| RHOA-GTP | 24 | 8/ND | 3/8 | 4 |
| Her2 | 6 | 5/10 | ND | 3 |

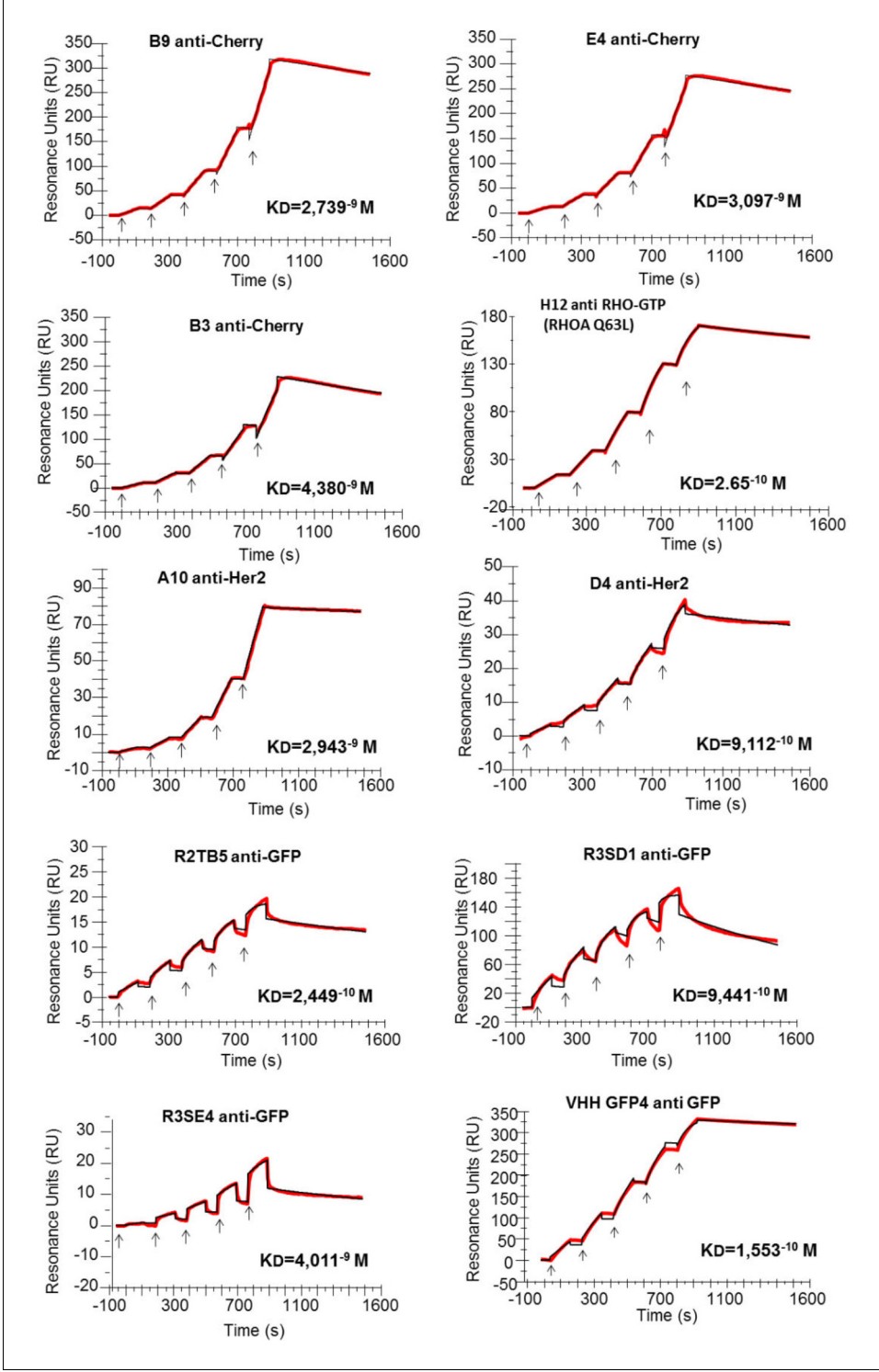

**Figure 6.** Affinity determination. Single cycle kinetics analysis was simultaneously performed on immobilized His fusion VHH antibodies (250–300 RU), with five injections of analytes (EGFP, HER2, RHOAQ63L and mCherry) at 3.125 nM, 6.25 nM, 12.5 nM, 25 nM, and 50 nM. Analytes injections lasted for 120 s each and were separated by 10 s dissociation phases. At this time of buffer exchange, a slight refraction index discrepancy between the sample and the flow buffer can induce a drop in resonance unit. This common bulk effect, which is clearly visible on sensorgrams with a smaller scale range on the RU axis (ie: R3SE4, R2TB5), does not affect the measurement of off-rate constant. Off-rate constant was calculated from an extended dissociation period of 10 min following the last injection according to the single cycle kinetics method. Each sensorgram (expressed in RUs as a function of time in

*Figure 6 continued on next page*

*Figure 6 continued*

seconds) represents a differential response where the response on an empty reference channel (Fc1) was subtracted. The red curves correspond to the data and the black curves represent the fit done by the BIAevaluation software. Note that the fitted curve is almost identical to the data curve in some cases like for example the RHOA Q63L or the HER2 binding measurement.

*Nizak et al., 2003a*) we then rationally designed CDR diversity with fixed CDR1 and CDR2 size and four CDR3 sizes (9, 12, 15 or and 18 amino acids). The power of modern gene synthesis approach permits to reach very high genetic diversity while controlling codon bias and cloning features. Fully random codon combination using NNN or NNK trinucleotide cannot prevent stop codon, undesired cysteine or hydrophobic residues to be incorporated, and it does not lead to the controlled probability of amino acid occurrence at a given position. Therefore a more rational design was implemented with defined set of codons for each CDR amino acid position so that it does not mimic natural diversity, in contrast to recently developed Fab synthetic libraries (*Prassler et al., 2011*; *Zhai et al., 2011*), but is rather optimized for intrinsic hydrophilicity or solubility. After a large-scale cloning of synthetic fragments, 3 billion independent clones were transformed in the bacteria. Library quality was confirmed by Sanger and next generation sequencing.

The library was validated by screening against various targets and in each case specific and highly functional antibodies were obtained (*Table 1*). Various selection schemes yielded a large diversity of high affinity and high selectivity binders. Selections were carried out using purified antigen coated on polystyrene, on magnetic beads or directly on the cell surface. In many cases, two rounds of selection were sufficient to obtain selective binders. We usually analyzed only 80 randomly picked clones because the diversity of specific binders was systemically high. Only a few selections led to antibodies usable in western blotting (anti-actin, ant-tubulin) probably because most screenings were done using

**Table 2.** Binding affinities of 9 selected hs2dAb fused to a 6HIS tag measured by surface plasmon resonance single cycle kinetics method. Dissociation equilibrium constant KD corresponds to the ratio between off-rate and on-rate kinetic constant $K_{off}/K_{on}$. Non relevant hs2dAb were used as negative controls and gave no detectable binding signal. A positive control endowed with subnanomolar affinity, the GFP binder VHH-GFP4, was analyzed in parallel to the GFP hs2dAbs. A KD of $1.55^{-10}$ M was measured for VHH-GFP4 which is similar to published values. The binding properties of the conformational H12 hs2dAb to the GTP loaded RHOA subfamily were measured using the L63 or L61 constitutively active mutants of RHO, RHOB, RHOC, RAC1 and CDC42 related small GTPases, as well as the negative mutant T19N of RHOA. ('no' means no detectable binding).

| hs2dAb-6xHis | Antigen | $k_{on}$ (M$^{-1}$ s$^{-1}$) | $k_{off}$ (s$^{-1}$) | KD(M) |
|---|---|---|---|---|
| R2TB5 anti-GFP | GFP | $1.24\ 10^{+6}$ | $3.05\ 10^{-4}$ | **$2.45\ 10^{-10}$** |
| R3SD1 anti-GFP | GFP | $7.07\ 10^{+5}$ | $6.68\ 10^{-4}$ | **$9.44\ 10^{-10}$** |
| R3SE4 anti-GFP | GFP | $1.45\ 10^{+5}$ | $5.83\ 10^{-4}$ | **$4.01\ 10^{-9}$** |
| Llama VHH GFP4 | GFP | $2.99\ 10^{+5}$ | $4.65\ 10^{-5}$ | **$1.55\ 10^{-10}$** |
| D4 anti-Her2 | Her2 | $1.79\ 10^{+5}$ | $1.63\ 10^{-4}$ | **$9.11\ 10^{-10}$** |
| A10 anti-Her2 | Her2 | $1.66\ 10^{+4}$ | $4.88\ 10^{-5}$ | **$2.94\ 10^{-9}$** |
| B9 anti-Cherry | mCherry | $6.14\ 10^{+4}$ | $1.68\ 10^{-4}$ | **$2.74\ 10^{-9}$** |
| E4 anti-Cherry | mCherry | $6.57\ 10^{+4}$ | $2.03\ 10^{-4}$ | **$3.10\ 10^{-9}$** |
| B3 anti-Cherry | mCherry | $6.19\ 10^{+4}$ | $2.71\ 10^{-4}$ | **$4.38\ 10^{-9}$** |
| H12 anti-RHO.GTP | RHOA Q63L | $4.81\ 10^{+5}$ | $1.28^{-4}$ | **$2.65\ 10^{-10}$** |
| H12 anti-RHO.GTP | RHOB Q63L | $2.24\ 10^{+5}$ | $3.59^{-4}$ | **$1.57\ 10^{-9}$** |
| H12 anti-RHO.GTP | RHOC Q63L | $1.12\ 10^{+6}$ | $5.41^{-5}$ | **$4.79\ 10^{-11}$** |
| H12 anti-RHO.GTP | RHOA T19N | no | no | no |
| H12 anti-RHO.GTP | RAC1 Q61L | $7.53\ 10^{+5}$ | $2.55^{-4}$ | **$3.3\ 10^{-10}$** |
| H12 anti-RHO.GTP | CDC42 Q61L | no | no | no |

natively folded targets. Accordingly, selected hs2dAb performed very well in other conventional immunoassays like ELISA, FACS, immunoprecipitation or immunofluorescence. Affinity measurements done by surface plasmon resonance revealed KD values in the order of 10 nanomolar and up to 50 picomolar. Such high affinities are rather good and usually rarely observed for monovalent binders obtained without in vivo immunization or in vitro affinity maturation steps (*Figure 6*; *Table 2*).

The NaLi-H1 library thus enables the rapid selection of diverse and highly functional binders. Because it is a fully synthetic, non immune, library, it does not depend on animal experimentation, it is not limited by natural immunogenicity or toxicity of antigens and allows to develop and adjust the selection without ethic consideration. In addition, because all steps are carried out in vitro, conditions can be tightly controlled. This allowed to develop powerful differential selection and to identify conformation-specific antibodies. This also allowed to directly screen for antibodies directed against antigens specifically present at the surface of a particular cell type. Such a differential selection will be a powerful approach to identify novel antigen at the surface of tumor or infected cells. Such antibodies may also represent powerful tools for diagnostic and therapeutic applications to target cells in human pathologies. For example, after dimerization using Fc domains, hs2dAb antibodies may be used to target tumor cells and benefit from antibody-dependent cell-mediated cytotoxicity (ADCC) for example. They may also be used directly as the smallest antibody-derived domain naked as an agonist or antagonist or armed for enhanced toxicity. Similarly, it may be labeled using radioactive compounds (e.g. $^{99}$mTc, $^{111}$In, $^{64}$Cu) and used to image tumors in patient using positron emission tomography. Altogether, he NaLi-H1 library may accelerate the identification of novel potent tools to be used in human clinical applications.

The synthetic scaffold we defined was based on the selection of a set of VHH able to fold properly in the bacteria cytosol. The goal was not only to define a robust scaffold that would be efficiently produced without aggregation but also to allow frequent selection of functional intrabodies. Intrabodies have been isolated from various antibody libraries (*Nizak et al., 2003a*; *Rothbauer et al., 2006*; *Tanaka and Rabbitts, 2010*; *Vercruysse et al., 2010*) as well as other protein scaffold like Darpin (*Tamaskovic et al., 2012*) or FN3 (*Koide et al., 2012*) which are devoid of cysteine. A peculiar feature of the NaLi-H1 library is that it is based on a humanized nanobody-like robust scaffold, stable in a reducing environment, while it still contains the two canonical cysteine residues. Stabilized nanobodies, human single domain scaffolds (*Christ et al., 2007*; *Saerens et al., 2005*) and libraries (*Goldman et al., 2006*; *Mandrup et al., 2013*) were reported before. However, to our knowledge, no synthetic library producing at high frequency functional intrabodies was developed before based on such stabilized scaffolds. Almost every hs2dAb antibodies we expressed in mammalian cytosol showed no sign of aggregation, which further supported the idea that the synthetic scaffold we designed is robust and highly resistant to reduction. Previous studies showed that functional intrabody identification often relied on additional steps of selection in a protein–protein interaction reporter system such as PCA (*Koch et al., 2006*) or bacterial 2 hybrid (*Pellis et al., 2012*), yeast IACT (*Tanaka and Rabbitts, 2010*) or F2H assays (*Zolghadr et al., 2008*). Using the NaLi-H1 library, we observed that without using particular selection schemes, functional intrabodies were frequently obtained. Although we did not formally compare the NaLi-H1 library to previous llama naïve or semi synthetic libraries, the functionality of selected hs2dAb was compared to a sub-nanomolar affinity intrabody, the GFP4 nanobody, which has been extensively used (*Caussinus et al., 2011*; *Kirchhofer et al., 2010*). We observed by monitoring its signal-to-noise ratio and by using it in a protein knockdown assay that NaLi-H1 can provide highly functional hs2dAb which appeared as good as intrabodies from immune libraries.

Intrabodies can be used in several applications like tracking of intracellular dynamics of endogenous proteins (*Nizak et al., 2003a*, *2003b*; *Rothbauer et al., 2006*) but the most appealing application is to use them for rapid protein inactivation in living cells. Intrabodies may be used to directly block their target proteins in cells. One of the conformation-sensitive antibody that was selected can be used to inhibit active RHO GTPase signaling in living cells and is as potent as the C3 exoenzyme toxin. Only few intrabodies have been described to be intrinsically inhibitors of protein activity (*Haque et al., 2011*; *Vercruysse et al., 2010*), and our results suggest that the NaLi-H1 library may enable rapid selection of inhibitory antibodies. The next challenge will be to select conformational sensors specifically directed against a particular member of closely related RHO subtypes (RHOA/B/C) which share more than 90% similarity in primary sequence. But in any case, our results show that the NaLi-H1 library allows the selection of efficient, conformation-specific, inhibitory intrabodies.

Another way intrabodies may be used to inactivate their targets in living cells is to fuse the intrabody to a dominant inhibitory domain. Following an idea pioneered by Affolter and colleagues (*Caussinus et al., 2011*), we showed here that intrabodies selected from the NaLi-H1 library can be fused to a proteasome-targeting domain to impose the specific degradation of their respective targets. This protein interference approach was validated using anti-EGFP hs2dAb and we believe that this approach will bring disruptive tools to generate rapid protein knockdown both in cell culture and in the animal.

In summary, we have designed a novel nanobody scaffold endowed with improved stability and created a highly diverse library, the NaLi-H1 library, that was successfully screened to identify highly functional binders directed against very diverse targets. We believe that this library will allow the fast, and fully in vitro, identification of immunological tools usable both for fundamental and medical applications.

## Materials and methods

### Plasmids and cloning

Artificial gene synthesis (Mr Gene, GmbH, Germany) composed of a 6His-Tag and a triple c-myc Tag was inserted into the pHEN2 phagemid vector (Griffin 1. library) between NotI and BamHI sites. CcdB gene from pENTR4 vector (Invitrogen - ThermoFisher Scientific, France) was inserted into the pHEN2 vector between NcoI and NotI sites. This vector allows to express antibody fragments in fusion, upstream, with the pelB leader to drive secretion in the periplasm and downstream with the PIII protein of M13 phages. An amber stop codon is present between the antibody and the pIII. This stop codon is partially suppressed in SupE *E. coli*. For expression and purification of dimeric antibodies, hs2dAb were inserted in vectors derived from pFuse (Invivogen, France) as described in *Moutel et al. (2009)*. For intrabody expression in mammalian cells, hs2dAb were digested by NcoI and NotI and ligated into the pIb-mEGFP, pEGFP or the pmCherry vectors (Clontech - Takara, USA). (See the Appendix for more details).

### CAT filter assay

Previously selected VHHs from naïve or immune libraries were subcloned into pAOCAT (*Monegal et al., 2012*) using the NcoI and NotI restriction sites. Chloramphenicol resistance assay was performed using BL21 (DE3) cells transformed with the pAOCAT-VHH fusion constructs. (See the Appendix for more details).

### Library construction

Details about the construction of the library can be found in the Appendix. In short, a synthetic design was ordered based on a statistic analysis of the diversity found in natural VHH and aiming at reducing hydrophobicity at some position. The size of the CDR1 and CR2 was fixed at 7 amino acids while 4 sizes of CDR3 were chosen (9, 12, 15 and 18 amino acids). Large -scale PCR was then carried out ensuring that at least $10^{10}$ DNA molecules were used as a matrix. Fragments were then cut and inserted into the pHEN2-3myc plasmid. The ligated DNA material was used to transform electro-competent *E. coli* TG1 cells (Lucigen Corp., Middleton, United States). Serial dilution was used to count the total number of bacteria transformed. A potential diversity of $3 \times 10^9$ was calculated. Transformed bacteria plated on 430 2xYT-ampicillin agar dishes (140 mm), grown overnight at 37°C, scrapped and stored in 30% of glycerol at −80°C.

### Ion torrent sequencing

IonTorrent sequencing library was prepared with the Ion Plus Fragment Library kit for AB Library Builder System (Life Technologies - TermoFisher Scientific, France) following manufacturer's instructions and was controlled on the Agilent 2100 Bioanalyzer (Agilent Technologies, France) with the High Sensitivity DNA Kit (Agilent Technologies). The sequencing template was prepared by emulsion PCR with the Ion OneTouch 2 system and the Ion PGM Template OT2 400 Kit (Life Technologies). Sequencing was performed on a IonTorrent Personal Genome Machine using the Ion PGM Sequencing 400 Kit and a 314v2 Ion chip (Life Technologies).

## Antigens

Human βActin was purchased from Sigma-Aldrich (France). RHOA GTPase fused to either an amino terminal Chitin Binding Domain or a streptactin binding peptide were produced in HEK293 cells. EGFP (as mCherry) in fusion with a streptavidine binding peptide (SBP) were produced through in vitro translation system (Roche Life Science, France) and used directly for screening without the need for purification.

Biotinylated Tubulin was purchased from Cytoskeleton, Inc. (Denver, United States). For p53, the 83 first amino acids of the NP_000537.3 isoform were produced in bacteria with a SNAP and His Tag, purified using Talon resin (Takara - Clontech) and biotinylated in vitro. HP1α was produced in bacteria with an avitag and a His Tag, and purified using Talon resin.

For HER2, the natural receptor was used as membrane protein target on SKBR3 cells.

For more details see the Appendix for more details.

## Phage display selections

Screening for ßactin was performed by panning in immunotubes as described (*Marks et al., 1991*). Screening for EGFP, Tubulin and p53 were performed in native condition as described (*Nizak et al., 2005*). Screening for HER2 was performed on surface cells as described (*Even-Desrumeaux and Chames, 2012*). Screening on RHO was performed in native condition. (See the Appendix for more details).

## Enzyme-linked immunosorbent assay (ELISA)

Individual clones were screened by monoclonal phage ELISA as described. (See the Appendix for more details).

## Western-blot

After boiling in SDS-PAGE loading buffer, the samples were separated on a 12% SDS-PAGE and transferred to nitrocellulose membranes (Whatman GmbH, Germany). Membranes were blocked in 3% non-fat milk-PBS with 0.2% Tween 20 for 1 hr at room temperature or overnight at 4°C. unpurified hs2dAb were used at 1/100 from culture supernatant and added to the membranes with an anti-hisTag antibody at 1/3000 (Sigma-Aldrich) for 90 min. Blots were then washed and incubated 1 hr with secondary anti-Mouse HRP labeled antibodies (diluted at 1/10000 in PBS 0.1% Tween 20) (Jakson ImmunoResearch Laboratories). After 5 washes with PBS 0.1% Tween 20, secondary antibodies were then revealed using the SuperSignal chemoluminescent reagent (Pierce) and Hyperfilm ECL (GE HealthCare). For RHO-GTP pull down, the primary anti RHOA mAb was used (Cell Signaling Technology; 1/1000). For protein knockdown experiments, 500 000 of transfected cells (mCherry positive cells) were sorted with a MoFlo Astrios flow cytometer (Beckman Coulter). Cells were lysed with SDS-Tris lysis buffer (Tris pH7.4 10 mM, SDS 1% supplemented with phosphatase and protease inhibitors). 20 μg of cell extracts were separated on 12.5% SDS-PAGE and electro transferred onto PVDF membranes. Blots were probed with a rabbit polyclonal anti-EGFP full length (Santa Cruz, sc-8334, 1:500), a mouse monoclonal anti-α-tubulin (Sigma, T5168, 1:25000) and an anti-myc HRP antibody (Novus Biologicals, NB600-341, 1:40000). Detection was performed using peroxydase conjugated secondary antibodies and Pierce ECL Western Blotting Substrate (Thermo Scientific Pierce).

## Immunofluorescence

Immunofluorescence screenings were performed on HeLa cells as described before (*Nizak et al., 2005*). (See the Appendix for more details).

## Transient transfection

Hela or HeLa S3 H2B-EGFP Cells cultured on coverslips were transfected according to the CaPO4 or jet prime procedure with 1 μg DNA per well (24 wells plate) or 10 μg DNA (10 cm2 diameter dish). Cells can be observed from 12 hr post-transfection on.

## Flow cytometry

For HER2 immunoassay, cell surface staining were performed in phosphate-buffered saline (PBS) supplemented with 1% SFV. 100 μL of supernatant (80 μL phages + 20 μL PBS/milk1%) were

incubated on $1.10^5$ cells for 1 hr on ice. Phage binding was detected by a 1:300 dilution of anti-M13 antibody (GE healthcare, France) for 1 hr on ice followed by a 1:1000 dilution of PE-conjugated anti-Mouse antibody (BD Bioscience, France) for 45 min. Samples were analyzed by flow cytometry on a FACSCalibur using CellQuest Pro software (BD Biosciences,France).

In the protein knockdown experiments, 48 hr after transfection, at least 10000 HeLa S3 H2B-GFP cells were analyzed on a MoFlo Astrios flow cytometer (Beckman Coulter France S.A.S) for their GFP fluorescence intensity. This fluorescence was analyzed in mCherry transfected cells and non trans-fected cells. Flow cytometry data were analyzed with Kaluza software (Beckman Coulter). 1 µM of proteasome inhibitor MG132 (Sigma-Aldrich) was used in the cell growth medium for 48 hr. Values reported represent median ± standard deviation (SD) of at least three independent experiments. p values were calculated with GraphPad Prism 6 (RRID:SCR_002798) using a Student's t test. **p<0.01; ***p<0.001; ****p<0.0001.

## Affinity measurement

All binding studies based on SPR technology were performed on BIAcore T200 optical biosensor instrument (RRID:SCR_008424, GE Healthcare). Capture of single domain Hs2dAb-6xHis was per-formed on a nitrilotriacetic acid (NTA) sensorchip in HBS-P+ buffer (10 mM Hepes pH 7.4, 150 mM NaCl, and 0.05% surfactant P20) (GE Healthcare). The four flow cells (FC) of the sensorchip were used: one (FC 1) to monitor nonspecific binding and to provide background corrections for analyses and the other three flow cells (FC 2, 3, and 4) containing immobilized Hs2dAb-6xHis for measurement.

For immobilization strategies, the four flow cells were loaded with nickel solution (10 µL/min for 60 s) in order to saturate the NTA surface with Ni2+ and an extra wash using running buffer contain-ing 3 mM EDTA after the nickel injection. Each His-tagged hs2dAb in running buffer was injected in flow cells at a flow-rate of 10 µL/min. The total amount of immobilized hs2dAb-6xHis was 250–300 resonance units. (RUs; 1 RU corresponds approximately to 1 pg/mm2 of protein on the sensor chip). A Single-Cycle Kinetics (SCK) analysis to determine association (on-rates), dissociation (off-rates) and affinity constants (kon, koff and KD respectively) was carried out. SCK method prevents potential inaccuracy due to sensorchip regeneration between cycles which are necessary in the conventional multiple cycle kinetics (MCK) (*Trutnau, 2006*). SCK binding parameters are evaluated for each injec-tion according to the tools and fit models of the BIAevaluation software, giving similar values than MCK. As hs2dAb were smaller proteins than their respective antigens, hs2dAbs were captured on the sensorchip while the recombinant antigens were used as analytes. Analytes were injected sequentially with increased concentrations ranging between 3.125 nM to 50 nM in a single cycle without regeneration of the sensorship between injections. Binding parameters were obtained by fit-ting the overlaid sensorgrams with the 1:1. Langmuir binding model of the BIAevaluation software version 1.0.

## Acknowledgements

The authors would like to acknowledge the help of the deep sequencing (ICGEx) and the imaging (PICT) platforms of the Institut Curie as well as the technical support of Aurélie Schneider, Selma Djender, and Alexis Arrial for the selection of anti-mCherry, anti-HER2 and anti-HP1α, Selma Djender and Anne Beugnet for the help in antibodies purification, Ahmed El Marjou for the production of recombinant HP1-Avitag and Anne-Laure Iscache for precious help in FACS experiments. We also would like to thank Yann Louault, Jean-Pierre Quivy and Christèle Maison for their help in character-izing anti-HP1 antibodies and Solene Hervé et Daniele Fachinetti for the characterization of the anti-p53 antbodies and the kind gift of the p53 shRNA expressing RPE-1 cell line.

## Additional information

### Competing interests

SM, FP: Co-inventor on a patent application (filled under ref: WO/2015/063331) that covers the hs2dAb scaffold and the commercial use of the library. The library has been licensed to Hybrigenics Service SA, which will perform screens on a fee-for-service basis. A consultant for Hybrigenics

Service SA. J-CR: Employed by, and a stockholder in, Hybrigenics Service SA. AO: Co-inventor on a patent application (filled under ref: WO/2015/063331) that covers the hs2dAb scaffold and the commercial use of the library. The library has been licensed to Hybrigenics Service SA, which will perform screens on a fee-for-service basis. The other authors declare that no competing interests exist.

### Funding

| Funder | Grant reference number | Author |
|---|---|---|
| Agence Nationale de la Recherche | ANR-09-BIOT-05 | Jean-Christophe Rain Franck Perez |
| Institut National de la Santé et de la Recherche Médicale | | Gilles Favre |
| Groupe de Recherche of the Claudius Regaud Institute | | Gilles Favre |
| Institut National de la Santé et de la Recherche Médicale | ITS-201103 | Franck Perez |
| IDEX Paris Sciences Lettres | ANR-10-IDEX-0001-02 PSL | Franck Perez |
| LABEX CellTisPhyBio | 11-LBX-0038 | Franck Perez |
| Centre National de la Recherche Scientifique | | Franck Perez |
| Institut Curie | | Franck Perez |
| Fondation pour la Recherche Médicale | DEQ20120323723 | Franck Perez |
| Aviesan | Project TAbIP | Franck Perez |

The funders had no role in study design, data collection and interpretation, or the decision to submit the work for publication.

### Author contributions

SM, Carried out most of the experiments leading to the library construction, Designed and set up some of the screening approaches, Analyzed the data, Prepared the figures, Wrote the manuscript; NB, Designed and set up some of the screening approaches and conduced the protein knockdown experiments, Analyzed the data, Prepared the corresponding figures, Corrected the manuscript; VB, Carried out deep sequencing, Designed and carried out the NGS experiments, Analyzed the data and helped prepare the figure; LK, Designed and set up the GTP-RHO experiments, Analyzed the data, Prepared the figure; EL, Set-up the conditions and carried out the time-lapse imaging of living cells, Acquisition of data, Analysis and interpretation of data; AdM, LL, Executed some biochemical experiments, Analyzed the data, Acquisition of data; J-CR, Helped for the construction of the library, Discussed the data, Conception and design, Acquisition of data, Analysis and interpretation of data; GF, Supported the study, Corrected the manuscript, Analysis and interpretation of data; AO, Contributed to the design and the supervision of the study, Analyzed the data, Acquisition of data, Wrote the manuscript; FP, Designed, supervised and supported the project, Wrote the article, Analysis and interpretation of data

### Author ORCIDs

Nicolas Bery, http://orcid.org/0000-0002-2643-3897
Laura Keller, http://orcid.org/0000-0002-1786-9760
Ario de Marco, http://orcid.org/0000-0001-7729-819X
Gilles Favre, http://orcid.org/0000-0002-2344-1883
Franck Perez, http://orcid.org/0000-0002-9129-9401

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

**Appendix**

## Supplementary methods

### Plasmids and cloning

myc-Tags synthetic gene inserted in pHen2 vector: NotI-HHHHHHGAAEQKLISEEDLNGAAEQKLISEEDLNGAAEQKLISEEDLNGAA*(tag)-pIII-BamHI.

p53: TP53 hMDM2 interacting domain (aa 1 to aa 83 ref NP_000537) has been cloned in pEB23, a pEMBL41 derivative in which MBP have been replaced by SNAP tag (New England Biolabs France). Fusion protein has been purified using Talon resin (Takara - Clontech) from 100 mL LB culture induced for 2 hr at 37°C with 1 mM IPTG. One µM of purified protein was biotinylated using 1 µM of SNAP-biotin (S9110S, New England Biolabs) for 2 hr at 25°C and purified by G50.

### Production of RHO GTPases amino terminal fusion construct.

Chitin binding domain (CBD) was amplified from pTYB2 (New England Biolabs) using the following oligos: CBDAgeIfw 5'-TACATTCAACCGGTCGCCACCATGACAAATCCTGGTGTATCCGCTTGG-3' and CBDTevRev 5'-CGAACTGAAAGTACAGATTCTCGCTGCCTTGAAGCTGCCA CAAGGCAGG-3'. RHOB cDNA was amplified from pGST-RHOBQ63L with the upstream primer TevAccRHOB Fw 5'-GAGAATCTGT ACTTTCAGTT CGGCGGTACC GCGGCCATCC GCAAGAAGCT GG-3'. A single step PCR created the CBDtevRHOB fusion product that was cloned in a pEYFP-RHOB (1) after digestion AgeI /BamHI, creating the pCBD-RHOB mammalian expression vector, RHOB was further exchanged by RHOAQ63L or RHOA T19N using Acc65I/BamHI cloning sites. The amino terminal CBD tag was replaced by either a dual Streptag (2S) or the superfolder GFP using AgeI and KpnI cloning sites. A 2S tag corresponding to a tandem repeat of the streptagII (IBA) separated by a flexible linker was gene synthesized (Geneart - ThermoFisher Biosciences). Superfolder GFP was amplified from pSuperfolder (gift from Cabantous S) using the AgeISFGFPFw 5'-AATGACATACCGGTCGCCACCATGGTGAGCAAGGGCGAGGAG-3' and SFGFPKpnIRev 5'-AAGTCAGGTACCGCCACCGCTGCCGCCACCCTTGTACAGCTCGTCCATGCCG-3'. The constructs pCBD-RHOA N19 or L63 were used in co-immunoprecipitation with GFP-hsdAb experiments, and the pSFGFP-RHOA N19 or L63 mutants were used to monitor H12 immunofluorescence experiments.

A CBS-2S dual tag was PCR amplified and cloned at the NotI site downstream of hs2dAb in pHEN6 vector, in order to produce pHEN- hs2dAb-CBD2S-6his. Purified hs2dAbs from this construct were used in RHOA pull down experiments.

### Construction of Fbox/NoFbox- hs2dAb-6Hismyc-IRES-MTS-FP expressing vectors.

A PCR was performed to extract the Fbox and noFbox sequences from the plasmids NSlmb-VHHGFP4 and NSnoFbox-VHHGFP4 using BspHIFboxFw (5' GTTCATGTCATGATGAAAAT GGAGACTG ACAAAATAATGG 3') and FboxNcoRev (5' CAAGATCCCATGGCGAGGTGGCGGCCAGTCCGCCAGTTG 3') primers. These fragments were inserted in the p-Ib plasmid (intrabodies expression vector with CMV promotor) by digesting with BspHI and NcoI. The vectors pIB-VHHGFP4, pIb-Fbox-VHHGFP4 and pIb-NoFboxVHHGFP4 were obtained. Then IRES-MTS-fluorescent protein was inserted by PCR downstream of the VHH. IRES was amplified from pCCEY (*Olichon et al., 2007b*) with insertion of flanking downstream restriction sites AgeI, NheI and Acc65I. Then a mitochondrial targeting sequence (MTS) from subunit VIII of human cytochrome c oxidase (pEYFP-MTS, Takara - Clontech) was inserted between NheI / AgeI, followed by mCherry between AgeI / Acc65I, thus creating the p-F-Ib_IRES_mito-mCherry; p-noF-Ib_IRES_mito-mCherry; p-Ib_IRES_mito-mCherry in which any hs2dAb can be inserted by NcoI and NotI cloning sites.

## CAT filter assay

Previously selected VHHs from naïve or immune libraries were subcloned into pAOCAT (*Monegal et al., 2012*) using the NcoI and NotI restriction sites. Chloramphenicol resistance assay was performed using BL21(DE3) cells transformed with the pAOCAT-VHH fusion constructs. Bacteria were used for inoculating 500 µL of LB containing kanamycin (35 µg/mL) and glucose (0.2%) and were grown at 37°C until $OD_{600}$ was 0.8. The cytoplasmic expression of the VHH-CAT-fusion proteins was induced for 2 hr by the addition of 0.2 mM IPTG. At the end of the induction period, bacteria aliquots of 4 µL were plated on LB-agar plates containing IPTG (0.1 mM) and increasing chloramphenicol concentrations ranging from 0 to 500 µg/mL. Bacteria were incubated at 30°C for 20 hr before quantification of the colony formation. The resistance level was evaluated according to the colony growth rate at the different chloramphenicol concentrations. Several VHHs that were giving colonies up to 500 µg/mL were compared to previously characterized intrabodies raised against GFP (nb GFP4) or Lamin (Lam) obtained by gene synthesis (GeneArt - ThermoFisher BioSciences) as well as to a thermostable VHH Re3. Liquid culture induced as above during 2 hr were diluted by serial dilution and 10 µL of each dilution were spotted on agar plates containing 250 µg/mL chloramphenicol (Cam) and 35 µg/mL kanamycin and incubated at 30°C for 20 hr. Colony were imaged using ChemiDoc MP imaging system (Biorad France).

The D10 clone was further subcloned into the pHEN6 expression vector, leading to a periplasmic expression higher than 5 mg/L of culture in E coli Xl1blue strain.

## Library construction

The amino acids residues chosen for each position of the synthetic CDR loops were determined after analyzing the natural diversity existing at each position on natural VHH and following the rules indicated bellow (see *Figure 1—figure supplement 3* for diversity comparison between natural VHH and hs2dAb)

at CDR1 position 1: Y, R, S, T, F, G, A, or D [Y,S, F were the most frequent amino acid]

at CDR1 position 2: Y, S, T, F, G, T, or T [T was the most frequent amino acid so we imposed a 3 fold higher stoichiometry for T at that position]

at CDR1 position 3: Y, S, S, S F, or W [S was the most frequent amino acid so we imposed a 3 fold higher stoichiometry for T at that position]

at CDR1 position 4: Y, R, S, T, F, G, A, W, D, E, K or N [natural diversity respected while avoiding hydrophobic residues]

at CDR1 position 5: S, T, F, G, A, W, D, E, N, I, H, R, Q, or L [natural diversity respected. Because the frequency of hydrophobic I was high at this position in natural VHH, some hydrophobic amino acids were accepted]

at CDR1 position 6: S, T, Y, D, or E [S, T, and Y were the most frequently found amino acids in natural VHH]

at CDR1 position 7: S, T, G, A, D, E, N, I, or V [a high frequency of hydrophobic I or V was observed in natural VHH but their proportion was reduced here by adding polar residues]

at CDR2 position 1: R, S, F, G, A, W, D, E, or Y [D and E were added to help for solubility and W for specific interaction in hydrophobic pockets]

at CDR2 position 2: S, T, F, G, A, W, D, E, N, H, R, Q, L or Y [introducing strong diversity]

at CDR2 position 3: S, T, F, G, A, W, D, E, N, H, Q, P [introducing strong diversity]

at CDR2 position 4: G, S, T, N, or D [G, S, T and N where the most frequent]

at CDR2 position 5: S, T, F, G, A, Y, D, E, N, I, H, R, Q, L, P, V, W, K or M [allowing full diversity]

at CDR2 position 6: S, T, F, G, A, Y, D, E, N, I, H, R, Q, L, P, V, W, or K [allowing strong diversity]

at CDR2 position 7: S, T, F, G, A, Y, D, E, N, I, H, R, Q, L, P, or V [allowing strong diversity]

at CDR3 for every position: S, T, F, G, A, Y, D, E, N, I, H, R, Q, L, P, V, W, K or M [introducing full diversity] CDR 3 length have been designed in order to produce 4 different subtype of hs2dAb with 9, 12, 15 or 18 residues in CDR3.

20 times, 1 µL (10 ng) of the synthesis (corresponding to $1.10^{10}$ molcules) were amplified by PCR in a total volume of 50 µL using 1 µL of Phusion DNA polymerase (New England Biolabs) with an equimolar mixture of the following primers:
 5'-AACATGCCATCACTCAGATTCTCG-3' and 5'-GTTAGTCCATATTCAGTATTATCG-3'
 PCR protocol consisted of an initial denaturation step at 98°C for 45 s followed by 20 cycles of 98°C for 10 s, 55°C for 30 s and 72°C for 30 s, and a final step extension at 72°C for 10 min. 7 × 150 µL of PCR were purified on 7 columns of a PCR clean-up kit (Macherey-Nagel). 55 µg of the resulting purified fragment of PCR and 80 µg of the pHEN2-ccdB-3myc phagemid were digested for 2 hr at 37°C with NcoI and NotI (New England Biolabs) in a total volume of 500 µL. A dephosphorylation step was added for the phagemid with a Calf intestinal alkaline phosphatase (Sigma-Aldrich) 30 min at 37°C. Digestions were purified on gel with respectively 4 and 6 columns of a gel extraction kit (Macherey-Nagel Sarl, France) in a final volume of 80 and 120 µL. Then, purified PCR fragment was ligated into pHEN2-ccdB-3myc, between the PelB leader signal and the pIII gene. 50 µg of phagemid and 19,2 µg of insert were ligated overnight at 16°C with 10 µL of high concentration T4 DNA ligase (New England Biolabs) in a total volume of 400 µL. Ligation was purified on 6 columns (Macherey-Nagel) with a total volume of 150 µL. The ligated DNA material was used to transform electrocompetent *E. coli* TG1 cells (Lucigen). 20 electroporations with 1 µL of ligation were performed according to the manufacturer's instructions (1800 V; 10 µF; 600 Ω). Each electroporation was resuspended with 1 mL of warm 2XYT, 1% glucose medium and incubated with a shaking agitation for 1 hr at 37°C. 380 mL of 2XYT, 1%glucose was added to the suspension and plated on 430 2xYT-ampicillin agar dishes (140 mm) overnight at 37°C. Library size was calculated by plating serial dilution aliquots. The colonies were scraped from the plates with liquid 2xTY and library was stored in the presence of 30% of glycerol at −80°C with 1 mL aliquots at OD = 38,4. $3.10^9$ individual recombinant clones were obtained.

## Phage display selections

Various methods outlined below were used to screen NaLi-H1 library. Three general advices can be given here: (1) In roughly 70% of the screen, specific clones were obtained. In many case, when not specific binders were recovered, we identified the antigen that was used as the principal cause of failure. It is often a good idea to insist and use an alternative target preparation and ensure its proper display during the selection phases. (2) A key step to obtain specific clones is the washing step. They should be extensive, numerous and in large volume, changing tubes several times. While it is always a good idea to start washing as quickly as possible, if the affinity of the hs2dAb obtained is too low then extended washing steps may be used in a subsequent screening to select for a low $k_{off}$. (3) We usually test for binders after the second and third rounds of selection, and more rarely after a fourth round. The method used to identify binders at the end of the selection should to be as close as possible to plan usage of the recombinant antibody. If immunofluorescence is planned, it ideally should be used to identify specific binders. If western blotting is the major application, It should be the preferred method to be used.

Immunotubes (Nunc) were coated with βActin at a concentration of 20 µg/mL in PBS overnight at 4°C. Immunotubes were then rinsed with PBS and blocked for 2 hr at room temperature with 2% w/v BSA or casein in PBS. After rinsing with PBS, > $10^{12}$ phage particles in 2% BSA or

casein were added to the immunotubes. The immunotubes were first incubated on shaker for 1 hr and then for 1 hr standing upright at room temperature. Unbound phage was washed away by rinsing the immunotubes fifteen times with PBS, 0.1% Tween 20 and five times with PBS. The bound phage fraction was eluted in 1 mL of 1 mg mL$^{-1}$ trypsin and inverting the tube for 10 min and then in 1 mL of 100 mM triethylamine and inverting the tube for 10 min. Triethylamine was neutralized by adding 0.5 mL 1 M Tris-HCl pH 7.4. The eluted phages were used for the infection of exponentially growing *E.coli* TG1.

Biotinylated antigens or SBP-antigen were diluted to obtain a 10–20 nM (1 mL final) and recovered on 50 µL streptavidin-coated magnetic beads (Dynal). As a reference, 10 nM of a 25-kDa protein like GFP represents 250 ng protein/mL (quantity used per round of selection). Efficient recovery was checked analyzing fractions of bound and unbound samples by Western blot using streptavidin-HRP or anti-AviTag antibodies. Antigen coated beads were incubated for 2 hr with the phage library (10$^{13}$ phages diluted in 1 mL of PBS + 0.1% Tween 20 + 2% non-fat milk). Phages were previously adsorbed on naked streptavidin-coated magnetic beads (to remove nonspecific binders). Phages bound to streptavidin-coated beads were recovered on a magnet and beads were washed 10 times (round 1) or 20 times (round 2 and 3) with PBS +Tween 0.1% on a magnet. Bound phages were eluted using triethylamine (TEA.100 mM) for 10 min and neutralized using 1 M Tris pH 7.4. The elution was done twice. *E. coli* (TG1) were infected with the eluted phages and plated on ampicillin containing agarose. For round 2 and round 3, only 10$^{12}$ phages were used as input.

Chitin binding domain from chitinase A1 (CBD) or 2strep tag (IBA) fusion of RHOA GTPase active mutant (RHOA$_{L63}$) were expressed transiently during 24 hr in HEK293 cells and captured freshly after cell lysis on magnetic beads before incubation with the library phages. Chitin magnetic beads ()New England Biolabs or streptactin coated magStrep HC (IBA Gmbh, Germany) were use in a similar way as described above with streptavidin beads. A phage display panning alternating rounds on chitin beads with rounds on streptactin beads was performed for 4 rounds. From the second round of panning, a depletion step on GDP loaded wild type RHOA or N19 inactive mutant was included.

When characterizing selected antibodies, various Non Relevant (NR) hs2dAb were used as controls. The one shown here are: A non-binding clone selected against FITC (in *Figure 5B–D* and *Figure 2—figure supplement 1*) and a non-binding clone selected against mCherry (in *Figure 5A* and *Figure 5—figure supplement 1*)

## Enzyme-linked immunosorbent assay (ELISA)

Briefly, individual clones from a master plate and harboring phagemids were inoculated into 250 µL of media (2xTY, 1% glucose, 100 µg mL$^{-1}$ ampicillin) and grown overnight at 37°C. Ten µL of this culture were inoculated into 250 µL of fresh media for 3 hr. The culture was infected with KM13 helper phage and grown overnight in 200 µL 2xTY medium supplemented with 100 µg mL$^{-1}$ of ampicillin and 75 µg mL$^{-1}$ of kanamycin at 25°C. Supernatants from these cultures were used in phage ELISA assays. Maxisorp ELISA plate were coated with 100 µL per well of protein antigen in 100 mM carbonate buffer pH 8.6 for 2 hr at room temperature (RT) or overnight (OVN) at 4°C. Usually, 1 µg to 10 µg of antigen were used per well. A control plate was coated with BSA. Wells were rinsed once with PBS Tween 0.1% and blocked with 200 µL per well of 2% Milk + PBS Tween 0.1% for 1 hr at RT or OVN 4°C. Wells were rinsed 5 times in PBS. Twenty µL of PBS Tween 0.1% Milk and 80 µL of the phage containing supernatant were added to each well in both phage ELISA plate (antigen-coated and BSA control) for 1 hr. Wells were washed 3 times with PBS Tween 0.1% and 3 times with PBS. HRP-anti-pIII antibody (GE Healthcare) was added at 1:5.000 in PBS Tween 0.1% + 2% milk and incubated for 40 min. Wells were washed 3 times with PBS Tween 0.1% and 3 times with PBS. Reaction was developed with 100 µL of HRP substrate solution (e.g. ABTS in citrate buffer) at RT for 15 min and the optic density was read at 405 nm. RHO-GTPases ELISA were performed as previously described.

## Cells

All cells are regularly tested for mycoplasma contamination. The HeLa cells (RRID:CVCL_0030) used in the study has been obtained from the B. Goud lab (Institut Curie). SNP characterisation has been carried out and may be shared upon request. The H2B-GFP cell line has been obtained from the lab of E. Nigg and has been described in *Silljé et al. (2006)*. RPE-1 immortalized by the expression of hTert (RRID:CVCL_4388) has well as the CHO cells have been obtained from the ATCC.

## Immunofluorescence and real-time imaging

HeLa cells were either fixed in 3% paraformaldehyde and permeabilized with PBS (plus 0.05% saponin and 0.2% BSA) or fixed and permeabilized with ice cold methanol for 4 min at −20°C. When needed, nocodazole (10 µM, 90 min) or cytochalasin D (5 µM, 60 min) were added before fixation. To activate p53 (*Figure 2—figure supplement 2*), cells were irradiated using a Stratagene UV crosslinker (100 joules/m$^2$) 24 hr before fixation. hs2dAb secreted in 600 µL 2xTY cultures (produced in 96-deep well plates) after overnight induction with 1 mM IPTG at 30°C were used undiluted without purification. hs2dAbs were co-incubated with 9E10 anti-myc monoclonal for 90 min on cells. Cells were then washed quickly twice and incubated with secondary antibodies for 30 min (Invitrogen - Thermofisher). The actin cytoskeleton was stained with Alexa594 phalloidin (1/1000).
Cells were grown on coverslip and transfected with F-construct 48 hr following the manufacturer's recommendation. Cells were fixed in 3.7% paraformaldehyde for 7 min then rinsed twice with PBS and mounted in Mowiol. Data acquisition was carried out on a Nikon Eclipse 90i (Nikon, France)and image processing with NIS Elements v3 software or using a Leica DM6000 B microscope equipped with a CoolSnap HQ2 camera controlled by Metamorph (Molecular Device Limited, UK, RRID:SCR_002368)

For time-lapse imaging, Hela cells seeded on glass the day before were transfected using calcium phosphate with mCherry-Rab6 and GFP-tagged anti-cherry expression plasmids. After 24 hr, medium is replaced by pre-warmed carbonate independent Leibovitz's medium (Invitrogen), and cells were imaged using an inverted microscope TiE (Nikon) equipped with spinning disk confocal head (PerkinElmer, Waltham, United States) and a CoolSnapHQ2 camera (Roper Scientific, France). Time-lapse images were acquired using a 63X objective and MetaMorph software (Molecular Device, RRID:SCR_002368) every 2 s for 4 min.

## Pull down

Pull down of endogenous RHO proteins loaded in active or inactive state were adapted from well-established GST-RBD pull down. HeLa cell (5.10$^6$ per immunoprecipitate) were lysed in buffer (50 mM Tris pH 7.4, 500 mM NaCl, 10 mM MgCl$_2$, 0.5% TritonX100). Crude protein lysates Extract was loaded with 0.2 mM GTPγS or 2 mM GDP in buffer supplemented with 10 mM EDTA for 30 min at 30°C. Reaction was stopped by adding 30 mM MgCl2. For immunoprecipitation, beads bound by freshly captured CBD tag hs2dAb (4 µg) or GST-RBD (40 µg) positive control were incubated with loaded protein suspension for 45 min at 4°C. Beads were washed 3 times with 50 mM Tris-HCl. pH 7.5. 150 mM NaCl. 10 mM MgCl2. 0.1% Tween20 and denatured in 2X Laemmli reducing sample buffer, boiled for 5 min and separated on 12.5% SDS-PAGE for Western Blot analysis with anti-RHOA followed by HRP-conjugated secondary antibodies.

Co-precipitations of intrabodies (myc- tagged hs2dAb) with RHOA CA active or DN inactive mutants were performed after transient co-transfection of pCBD-RHOAQ63L or pCBD-RHOAT19N with plb-myc in HeLa cells. After 24 hr, crude cell lysates containing CBD-RHO mutants in buffer (50 mM Tris pH 7.4. 150 mM NaCl / 10 mM MgCl2 / 0.5% TritonX100 / 10% glycerol) were incubated with chitin beads for 1 hr at 4°C. Co precipitation was revealed by RHOA antibody and myc antibody.

## Production and Purification of recombinant protein for affinity measurement

The monovalent hs2dAb were produced in bacterial periplasm. Briefly, XL1blue E.coli transformed with pHEN-hs2dAb-CBD-2S-6his or pHEN-Hs2dAB-6his were grown in TB-ampicillin (100 µg/mL) medium supplemented with 1% glucose in the growth phase and with 0.1% glucose during induction with 1 mM IPTG. After 16 hr of expression at 28°C, the cells were harvested and periplasmic proteins were extracted in TES (Tris 100 mM pH 8, EDTA 1 mM, Sucrose 500 mM) after osmotic shock.hs2dAb were purified in batch using IMAC affinity chromatography using Ni-NTA beads CompleteHis resin (Roche Life Science) following manufacturer indications. Elution were dialyzed in PBS 10%glycerol.

Bivalent hs2dAb were produced as fusion proteins with the Fc domain of human IgG2 as described (*Moutel et al 2009*). hs2dAb were sub-cloned in pFuse-hIgG-Fc2 plasmid (NcoI/NotI) inframe between the interleukin-2 (IL2) secretion signal and the Fc domain. 4 days after transient transfection in CHO cells, supernatants were recovered and used directly for immunofluorescence staining of cells or for Western blot analysis. Hs2dAb-Fc were then revealed using anti-Human Fc fluorescent or HRP secondary antibodies (Jakson ImmunoResearch Laboratories, West Grove, United States).

GST-GFP or GST-mCherry were expressed in the cytosol by overnight induction in 1 liter of *E. coli* BL21 cells by addition of 100 µM IPTG. After centrifugation and freezing, the bacterial pellet was resuspended in 50 mM Tris-HCl, pH 8.5, 150 mM NaCl, 5 mM MgCl2, 1% Triton X-100, 10 mM dithiothreitol, 0.1 mg/ml DNase I, and protease inhibitors (Sigma-Aldrich). After centrifugation, the supernatant fraction was bound to 500 µl of pre-washed glutathione-Sepharose beads and purified in batch according to manufacturer instructions (GE Healthcare).

RHOA CA or DN mutants as well as RAC1 CA tagged with the twin Streptag II (IBA) were expressed in the cytosol of BL21 Star (DE3) pRARE *E.coli* cells from a pET vector and purified in the same buffer as GST-GFP but on the streptactin resin (IBA) according to manufacturer instructions.

