## [Decision Letter]

Thank you for submitting your article "NaLi-H1: A universal synthetic library of humanized nanobodies providing highly functional antibodies and intrabodies" for consideration by *eLife*. Your article has been favorably evaluated by Sean Morrison (Senior editor) and four reviewers, one of whom is a member of our Board of Reviewing Editors. The reviewers have opted to remain anonymous.

The reviewers have discussed the reviews with one another and the Reviewing Editor has drafted this decision to help you prepare a revised submission.

Summary:

This paper describing the generation of a synthetic nanobody library with an optimized humanized scaffold, illustrates its usefulness by showing examples of successful selection of antibodies against different antigens, including conformation-specific nanobodies, and demonstrates the use of some of these antibodies as intrabodies for protein detection and degradation. Previously, immunization of Camelids was necessary to isolate functional nanobodies. With the NaLi-H1 library this immunization step is removed and the screening is entirely performed in vitro, where at each step the conditions can be controlled and manipulated. This has several advantages as no animal experimentation is needed, and binders to highly conserved or toxic antigens can be isolated. The provided data suggest that the library will be a useful and versatile tool for selection of various nanobodies. Some of the described antibodies, such as the conformation-specific nanobody against active Rho, will be directly useful for the cell-biological community.

Essential revisions:

1) The library is assessed for diversity by sequencing, but not at all for functionality in terms of nanobody expression, folding and solubility. Given that the library only partially attempts to mimic natural diversity, reducing the levels of hydrophobic amino acids, there is a danger that attempts to improve upon nature may lead to increased aggregation/poor folding etc. Consequently, it would be important to assess the levels of functional expression of 24-48 random clones picked from the library. This could be done by assessing the levels of soluble expression levels, or preferably, using protein A (if protein A binds to the scaffold, given that it is VH3-like) to assess the percentage of protein A binding clones – which would provide an assessment of correct folding.

2) The staining of endogenous proteins with and without knockouts mediated by siRNA or Crispr-CAS9 should be demonstrated for the original anti-lamin nanobodies using the difficult scaffolds, as well as subsequently selected nanobodies. This will show first that the nanobodies are functional against natural proteins at natural levels, and second that there is no off-target binding (if knockout eliminates staining). Furthermore, functional positive control antibodies should also be included as comparisons.

3) Wherever the phrase "data not shown" is used, either the data should be shown (in the supplementary), or the phrase should be eliminated. Supplementary files attached to primary figures allow all data to be shown.

4) Where western blots are shown (e.g. Figure 3) show the whole blot, not just that portion { ± } 5KDa around the expected target size.

5) During the description of the library the authors mention () that the purified *sdAb^D10^* shows stability after heat treatment. They support this observation in Figure 1—figure supplement 1 panel B. One of the advantages of nanobody is their heat stability so the heat-purification experiment at 70 °C is not sufficient to support the claim that such scaffold is a better "heat-resistant" nanobody. This claim should be better supported or removed.

6) Please explain in a bit more detail how the diversity in the CDR3 was introduced, especially in the Materials and methods section.

7).Since this publication is mainly supposed to be a method-based, resource paper, it would greatly benefit from an extended description of a general protocol outline for the isolation of binders (starting from the preparation of the antigen, to the phage display and to the final screening method, with a few tips and tricks at the crucial steps). Researchers that want to use the NaLi-H1 library could use this as a starting point.

8) The authors present the success stories, but it would be good to know what percentage of attempted antibody screens was not successful.

9) Please make sure that a proper disclaimer regarding your financial interest in the commercial distribution of the library is provided in the published version of the paper.

---

## [Author Response]

*Essential revisions:*

*1) The library is assessed for diversity by sequencing, but not at all for functionality in terms of nanobody expression, folding and solubility. Given that the library only partially attempts to mimic natural diversity, reducing the levels of hydrophobic amino acids, there is a danger that attempts to improve upon nature may lead to increased aggregation/poor folding etc. Consequently, it would be important to assess the levels of functional expression of 24-48 random clones picked from the library. This could be done by assessing the levels of soluble expression levels, or preferably, using protein A (if protein A binds to the scaffold, given that it is VH3-like) to assess the percentage of protein A binding clones – which would provide an assessment of correct folding.*

It is a good question although not easy to solve. Assessing the folding of non-binding clones is challenging. Although the binding to Protein A and Protein L has indeed been described for some VH3 (Potter, Li and Capra.J Immunol 1996; 157:2982-2988; Rodrigo, Gruvegård and Van Alstine. Antibodies 2015, 4, 259-277), the hs2dAb scaffold cannot bind Protein A nor Protein L. We believe that the synthetic scaffold does not preserve the binding interface of some natural VH3 and mainly does not contain the residues in CDR2 that determine this interaction. Alternatively, to assess the relative folding of hs2dAb, we picked randomly 26 clones and carried out sequential ultracentrifugation. We also analyzed the behavior of a few selected clones directed against GFP, mCherry or lamin, as well as the natural anti-GFP nanobody GFP (also known as Chromobody). All the antibodies we tested were found (~100% ) in the supernatant after a 250 000 g spin. After warming at 90°C for 20 minutes, over 70% stays in the supernatant confirming strong resistance to denaturation. We now provide these data as Figure 1—figure supplement 4.

*2) The staining of endogenous proteins with and without knockouts mediated by siRNA or Crispr-CAS9 should be demonstrated for the original anti-lamin nanobodies using the difficult scaffolds, as well as subsequently selected nanobodies. This will show first that the nanobodies are functional against natural proteins at natural levels, and second that there is no off-target binding (if knockout eliminates staining). Furthermore, functional positive control antibodies should also be included as comparisons.*

A similar point was raised by Reviewer #2. The specificity of the anti-Lamin antibody is quite clear as it is simply a CDR grafting from a published antibody and we obtained the same staining. However, we recognize that we should do a better job showing the specificity of some of the selected binders. We added an shRNA + UV treatment experiment for the anti-p53 (new Figure 2—figure supplement 2). Using siRNA was more complicated for HP1 because of inter- dependence that exist between the different isoform of HP1. As it would be beyond the scope of this article to study HP1 we decided not to provide siRNA data. Instead we carried out immunofluorescence upon overexpression of the different isoforms as well as western blotting (new Figure 2—figure supplement 3). These experiments suggest that the selected HP1 antibody is directed against α, β and γ HP1. We also carried out nocodazole and cytochalasin D experiments to characterize better the anti-tubulin and anti-actin hs2dAb, respectively (new Figure 2—figure supplement 1).

*3) Wherever the phrase "data not shown" is used, either the data should be shown (in the supplementary), or the phrase should be eliminated. Supplementary files attached to primary figures allow all data to be shown.*

We removed the data not shown statements. In particular, we now show control experiments for p53 and HP1 antibodies.

*4) Where western blots are shown (e.g. Figure 3) show the whole blot, not just that portion { ± } 5KDa around the expected arget size.*

We already provided the full blot when we tested our novel anti-tubulin and actin antibodies because it was important in this experiment to see the potential background. For the pull down experiments (Figure 3) we thought that it would be less important because what we score is whether the protein is pulled-down or not; such an experiments could be done by dot blot with essentially the same precision. In any case, we now provide the full, un-cropped, blots as supplementary data (new Figure 3—figure supplement 1, new Figure 5—figure supplement 3, new Figure 5—figure supplement 4). Note that we performed a new set of pull-down assays (Figure 3—figure supplement 1) because previous un-cropped blots were slightly dirty.

*5) During the description of the library the authors mention () that the purified sdAb^D10^ shows stability after heat treatment. They support this observation in Figure 1—figure supplement 1 panel B. One of the advantages of nanobody is their heat stability so the heat-purification experiment at 70 °C is not sufficient to support the claim that such scaffold is a better "heat-resistant" nanobody. This claim should be better supported or removed.*

We indeed did not test extensively heat resistance of the new scaffold. However, our conclusions were still mild ("Purified *sdAb_D10_* showed excellent solubility, stability after heat treatment*…*") but it is true that only a 70 °C treatment was used. We now have tested a 90 °C treatment and confirmed partial resistance to heat denaturation. We however clarified our conclusions by removing “heat-resistant” and we now indicate that "*sdAb_D10_*showed excellent solubility, stability after treatment at 70 °C and we did not observe aggregation". In the Discussion we now mention that the hs2dAb displays "the hs2dAb displays partial resistance and/or refolding after treatment for 20 min at 90°C."

*6) Please explain in a bit more detail how the diversity in the CDR3 was introduced, especially in the Materials and methods section.*

More details are now provided in the Methods section. Because we thought that it may now be too long to be included in the main text, we now summarize the methods we used for library construction in the main material and methods section and provide more data in the Appendix.

*7) Since this publication is mainly supposed to be a method-based, resource paper, it would greatly benefit from an extended description of a general protocol outline for the isolation of binders (starting from the preparation of the antigen, to the phage display and to the final screening method, with a few tips and tricks at the crucial steps). Researchers that want to use the NaLi-H1 library could use this as a starting point.*

We used a variety of protocols in this manuscript that we outlined in the previous version. We have now added a sort of generic "tips and tricks" summary as an introduction to the screening methods.

*8) The authors present the success stories, but it would be good to know what percentage of attempted antibody screens was not successful.*

This is an interesting data indeed but not simple to provide because in some case, the screen fails at first but then, tuning the conditions, it finally succeeds. Probably more than 70% of the screens eventually work. When they failed, we generally identified the antigen as being the main cause. We now indicate these points in the generic "tips and tricks" header (see point #7).

*9) Please make sure that a proper disclaimer regarding your financial interest in the commercial distribution of the library is provided in the published version of the paper.*

We indicated this in the manuscript (Competing financial interests). We shall see with the editors where is the proper section to indicate this point. Looking at Instruction to Authors, we realized that it should probably be present on the first page. Also, we now indicate the WIPO number.